# Joint Generative Modeling of Grounded Scene Graphs and Images via Diffusion Models

**Bicheng Xu**[*]                                        *bichengx@cs.ubc.ca*
*University of British Columbia*
*Vector Institute for AI*

**Qi Yan**[*]                                            *qi.yan@ece.ubc.ca*
*University of British Columbia*
*Vector Institute for AI*

**Renjie Liao**                                          *rjliao@ece.ubc.ca*
*University of British Columbia*
*Vector Institute for AI*
*Canada CIFAR AI Chair*

**Lele Wang**                                            *lelewang@ece.ubc.ca*
*University of British Columbia*

**Leonid Sigal**                                         *lsigal@cs.ubc.ca*
*University of British Columbia*
*Vector Institute for AI*
*Canada CIFAR AI Chair*

**Reviewed on OpenReview:** *https://openreview.net/forum?id=2cxxZI2LOL*

## Abstract

A grounded scene graph represents a visual scene as a graph, where nodes denote objects (including labels and spatial locations) and directed edges encode relations among them. In this paper, we introduce a novel framework for joint grounded scene graph - image generation, a challenging task involving high-dimensional, multi-modal structured data. To effectively model this complex joint distribution, we adopt a factorized approach: first generating a grounded scene graph, followed by image generation conditioned on the generated grounded scene graph. While conditional image generation has been widely explored in the literature, our primary focus is on the generation of grounded scene graphs from noise, which provides efficient and interpretable control over the image generation process. This task requires generating plausible grounded scene graphs with heterogeneous attributes for both nodes (objects) and edges (relations among objects), encompassing continuous attributes (*e.g.*, object bounding boxes) and discrete attributes (*e.g.*, object and relation categories). To address this challenge, we introduce DiffuseSG, a novel diffusion model that jointly models the heterogeneous node and edge attributes. We explore different encoding strategies to effectively handle the categorical data. Leveraging a graph transformer as the denoiser, DiffuseSG progressively refines grounded scene graph representations in a continuous space before discretizing them to generate structured outputs. Additionally, we introduce an IoU-based regularization term to enhance empirical performance. Our model outperforms existing methods in grounded scene graph generation on the Visual Genome and COCO-Stuff datasets, excelling in both standard and newly introduced metrics that more accurately capture the task's complexity. Furthermore, we demonstrate the broader applicability of DiffuseSG in two important downstream tasks: (1) achieving superior results in a range of

---

[*]Equal contribution

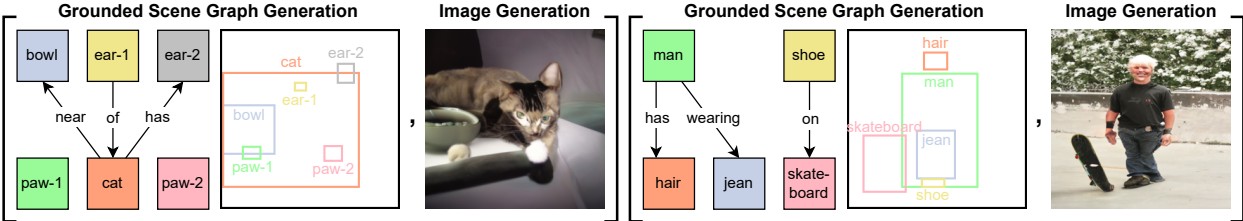

Figure 1: **Joint grounded scene graph and image modeling.** We model the joint distribution of grounded scene graph - image pairs via two steps: first training our proposed DiffuseSG model to produce grounded scene graphs and then utilizing a conditional image generation model to generate images. LayoutDiffusion (Zheng et al., 2023b) is used to generate images in these examples. Results shown are sampled from models trained on the Visual Genome dataset (Krishna et al., 2017).

grounded scene graph completion tasks, and (2) enhancing grounded scene graph detection models by leveraging additional training samples generated by DiffuseSG. Code is available at `https://github.com/ubc-vision/DiffuseSG`.

# 1 Introduction

A grounded scene graph is a graph-based representation that captures semantics of the visual scene, where nodes correspond to the objects (including their identity/labels and spatial locations) and directed edges correspond to the spatial and functional relations between pairs of objects. Grounded scene graphs have been widely adopted in a variety of high-level tasks, including image captioning (Yang et al., 2019; Zhong et al., 2020) and visual question answering (Damodaran et al., 2021; Qian et al., 2022). Various models (Kundu & Aakur, 2023; Jung et al., 2023; Zheng et al., 2023a; Jin et al., 2023; Biswas & Ji, 2023; Li et al., 2024; Hayder & He, 2024) have been proposed to detect grounded scene graphs from images. Such models require supervised training with image - grounded scene graph pairs, which is costly to annotate.

Motivated by this and the recent successes of diffusion models, in this paper, we tackle the problem of joint generative modeling of grounded scene graphs and corresponding images via diffusion models. The benefits of such generative modeling would be multifaceted. First, it can be used to generate synthetic training data to augment training of discriminative grounded scene graph detection approaches discussed above. Second, it can serve as a generative scene prior which can be tasked with visualizing likely configurations of objects in the scene conditioned on partial observations via diffusion guidance. For example, where is the likely position of the chair given placement of the table and sofa. Third, it can be used for controlled image generation, by allowing users to sample and edit grounded scene graphs, and then conditioned on them, generate corresponding images.

Diffusion models (Sohl-Dickstein et al., 2015; Ho et al., 2020; Song et al., 2021) have been shown to excel at modeling complex distributions, generating realistic high-resolution images (Saharia et al., 2022; Rombach et al., 2022; Nair et al., 2023; Ruiz et al., 2023; Epstein et al., 2023) and graphs (Jo et al., 2022; Vignac et al., 2022; Yan et al., 2023). However, joint generation of an image and the corresponding graph representation is challenging. To simplify the task, we factorize the joint distribution as a product of a grounded scene graph prior and a conditional distribution of images given a grounded scene graph. The conditional distribution has been widely studied in the form of layout-to-image generation (Zhao et al., 2019; Sun & Wu, 2019; 2021; Zheng et al., 2023b), a task of generating images based on spatial layouts that can be constructed from grounded scene graphs[1]. Therefore, in this paper we mainly focus on modeling the first term, *i.e.*, building a generative grounded scene graph model. This task in itself is challenging as it requires generation of graphs with heterogeneous attributes, *e.g.*, real-valued node attributes like object bounding boxes and categorical edge attributes like relation types.

---

[1]Though, notably, in the process, edge information is often not utilized.

Table 1: **Characterization of related, but often conflated, concepts encountered in the literature.** The representations used to describe the visual scene are: layout, scene graph, and grounded scene graph.

| Concept | Attributes Contained | | |
| --- | --- | --- | --- |
| | Object Label | Object Location | Relation Label |
| Layout | ✓ | ✓ | |
| Scene Graph | ✓ | | ✓ |
| Grounded Scene Graph | ✓ | ✓ | ✓ |

To generate grounded scene graphs, we propose DiffuseSG which is a diffusion-based generative model capable of generating plausible grounded scene graphs. To deal with heterogeneous attributes, we explore various encodings of categorical node/edge representations. We also design a graph transformer architecture that successively denoises the continuous graph representation which in the end produces clean grounded scene graph samples via simple discretization. Moreover, we introduce an intersection-over-union (IoU)-based training loss to better capture the distribution of bounding box locations and sizes. To generate images conditioned on grounded scene graphs, besides using one existing layout-to-image generation model, we also experiment with a relation-aware image generator built upon ControlNet (Zhang et al., 2023b).

In summary, our contributions are as follows. (1) We propose a novel joint grounded scene graph - image generation framework, by factorizing the joint distribution into a grounded scene graph prior and a conditional distribution of images given the grounded scene graph. In this context, we propose a diffusion-based model, named DiffuseSG, for grounded scene graph generation, which jointly models node attributes like object classes and bounding boxes, and edge attributes like object relations. Particularly, the technical novalties lie in the tensor representation of the grounded scene graph, separate prediction heads for node and edge attributes, an IoU-based training loss, and exploration among different encoding mechanisms for the categorical data. (2) We show that our model significantly outperforms existing unconditional scene graph generation models, layout generation models, and general-purpose graph generative models on both standard and newly introduced metrics that better measure the similarity between observed and generated grounded scene graphs. (3) We show that our model performs well on various grounded scene graph completion tasks using diffusion guidance. Moreover, paired with a conditional image generation model, our model generates grounded scene graph - image pairs which serve as extra training data for the downstream grounded scene graph detection task. The observed performance improvement highlights the practical significance of our joint modeling framework in real-world applications.

## 2 Representation Taxonomy

There are three similar but distinct visual scene representations that are often referenced in the literature. We formally define them here and in Tab. 1 to avoid conflating them. We strictly follow these definitions for the remainder of the paper.

**Layout.** Layouts comprise objects and their corresponding bounding boxes. There is no encoding of semantic relations between these objects beyond those implied by their placement. This representation is widely explored in the *layout generation* task, and utilized in the task of *image generation from layouts*.

**Scene Graph.** Scene graphs encode object labels and their corresponding relations in the form of directed graphs. As such, a scene graph can encode that a certain object (*e.g.*, a car) is *next to* another object, but it can not precisely localize those objects in a scene. This representation is mainly used in the *scene graph-to-image generation* task. More closely related to our task, some works generate scene graphs unconditionally.

**Grounded Scene Graph.** Grounded scene graphs can be viewed as a combination of layouts and scene graphs as defined above. A grounded scene graph is a directed graph where nodes correspond to objects encoded by labels and bounding boxes, and edges correspond to relations. This representation is extensively studied in the literature on *grounded scene graph detection from images*. In this paper, we propose DiffuseSG for the task of unconditional grounded scene graph generation.

## 3 Related Works

**Diffusion Models.** Diffusion models achieve great success in a variety of generation tasks nowadays, ranging from image generation (Kumari et al., 2023; Kim & Kim, 2024; Miao et al., 2024; Zeng et al., 2024b; Qu et al., 2024; Lin et al., 2024; Wei et al., 2024), video generation (Sun et al., 2023; Zeng et al., 2024a; Qing et al., 2024; Wang et al., 2024a; Skorokhodov et al., 2024; Liang et al., 2024; Gupta et al., 2024; Melnik et al., 2024; Liu et al., 2024b), text generation (Li et al., 2022b; Gong et al., 2022; Yuan et al., 2022; Dieleman et al., 2022; Ye et al., 2023; Lin et al., 2023; Wu et al., 2023), to simple graph generation (Jo et al., 2022; Vignac et al., 2022; Jo et al., 2023; Yan et al., 2023; Cho et al., 2024; Xu et al., 2024), *e.g.*, on molecule datasets (Irwin et al., 2012; Ramakrishnan et al., 2014), which usually have less than 10 node types and less than 5 edge types. These models contain two key processes: a forward process, which typically involves adding Gaussian noise to clean data, and a reverse denoising process that is often implemented using architectures such as U-Net (Ronneberger et al., 2015) or transformer (Vaswani et al., 2017). Since the grounded scene graph is fundamentally a graph structure, our proposed grounded scene graph generation model is conceptually similar to the ones for simple graph generation. However, our model goes beyond merely generating the graph structure of node (object) and edge (relation) labels. It also generates the object locations, in the form of bounding boxes. Also, our grounded scene graph data is more complex than the molecule data in terms of the numbers of node and edge types, *e.g.*, the Visual Genome dataset (Krishna et al., 2017) contains 150 node and 50 edge types. This diversity necessitates a re-evaluation of many design choices traditionally made in graph generation models.

**Layout Generation.** Layout generation focuses on creating image layouts, which comprise object labels and their corresponding bounding box locations. In contrast, our proposed grounded scene graph generation goes a step further by also generating the relations among these objects. Existing layout generation models typically take the form of VAE (Jyothi et al., 2019; Lee et al., 2020; Arroyo et al., 2021), GAN (Li et al., 2020a;b), transformer or BERT type language models (Gupta et al., 2021; Kikuchi et al., 2021; Kong et al., 2022; Jiang et al., 2023), or diffusion models (Inoue et al., 2023; Chai et al., 2023; Hui et al., 2023; Levi et al., 2023; Zhang et al., 2023a; Shabani et al., 2024). These layout generation models usually work on graphical layout generation problems, *e.g.*, designing layouts for mobile applications (Deka et al., 2017; Liu et al., 2018), documents (Zhong et al., 2019), or magazines (Zheng et al., 2019). The object bounding boxes of these layouts are expected to be well-aligned and not overlapping with each other. Thus, the layout generation models are usually measured on the alignment and intersection area of the generated bounding boxes. However, the object bounding boxes in our grounded scene graphs are naturally not aligned and usually occlusions occur. Therefore, we replace the evaluation metrics used in the layout generation literature with new ones that better capture characteristics of grounded scene graphs.

**Unconditional Scene Graph Generation.** Garg et al. (2021) introduce the task of unconditional scene graph generation, where a scene graph is generated from noise. They propose an autoregressive model for the generation, first sampling an initial object, then generating the scene graph in a sequence of steps. Each step generates one object node, followed by a sequence of edges connecting to the existing nodes. A follow-up work (Verma et al., 2022) proposes a variational autoencoder for this task, where a scene graph is viewed as a collection of star graphs. During generation, it first samples the pivot graph, and keeps adding star graphs to an existing set. The scene graph in these works is defined to only have object and relation category labels; object bounding box locations are omitted. Compared with these works, we use a diffusion-based model for unconditional grounded scene graph generation, where node and edge attributes are generated at the same time. We also include the object bounding box location in our grounded scene graph representation.

**Image Generation from Layouts or Scene Graphs.** There exist two conditional image generation tasks in the literature: layout-to-image (Zhao et al., 2019) and scene graph-to-image (Johnson et al., 2018). The conditions in the layout-to-image task are object labels and their bounding box locations, while the conditions in the task of scene graph-to-image are the object labels and relation labels. These tasks were initially widely explored within the GAN framework (Li et al., 2019; Ashual & Wolf, 2019; Sun & Wu, 2019; Dhamo et al., 2020; Li et al., 2021b; Sun & Wu, 2021; He et al., 2021; Sylvain et al., 2021). With the increasing popularity of diffusion models and their excellence at modeling complex distributions, diffusion models now become the default choice to accomplish these tasks (Yang et al., 2022; Cheng et al., 2023;

Zheng et al., 2023b; Farshad et al., 2023; Liu & Liu, 2024; Liu et al., 2024a; Wang et al., 2024b). However, the grounded scene graphs considered in our task contain object labels, object locations, and relation labels. These attributes can not be encoded altogether into any of the existing image generation models. This motivates us to build and explore diffusion-based relation-aware layout-to-image model.

# 4 Joint Grounded Scene Graph - Image Pair Modeling

We propose to model the joint distribution of grounded scene graphs and their corresponding images by modeling it as a product of two distributions. Denote the grounded scene graph by $S$ and the image by $I^2$. The joint distribution of grounded scene graph and image pairs can be factorized as $p(S, I) = p(S)p(I|S)$, from which one can easily draw samples in a two-step manner; first from the prior $p(S)$ and then from the conditional $p(I|S)$. Hence, we first build a grounded scene graph generation model to learn $p(S)$, *i.e.*, the underlying prior of grounded scene graphs. Second, we employ a conditional image generation model to capture $p(I|S)$, the conditional image distribution.

## 4.1 Grounded Scene Graph Generation

In this section, we first formally define our proposed grounded scene graph generation task. We then present our DiffuseSG model specifically designed for the task. DiffuseSG is a continuous diffusion model, whose training and sampling utilize the stochastic differential equation (SDE) formulation. We begin by providing some background on the SDE-based diffusion modeling, followed by an in-depth explanation of DiffuseSG.

### 4.1.1 Grounded Scene Graph Generation Task

A grounded scene graph $S$, consisting of $n$ nodes, can be described using node and edge tensors, denoted as $(V, E)$. We denote the space of node features by $\mathcal{V}$ and the space of edge features by $\mathcal{E}$, and the space of grounded scene graphs by $\mathcal{S} = \mathcal{V} \times \mathcal{E}$. The node tensor $V = [v_1; v_2; \ldots; v_n] \in \mathbb{R}^{n \times d_v}$, captures the node labels and their bounding box locations, where $d_v$ represents the dimension of the node feature. Each node feature $v_i = [c_i, b_i]$ combines a discrete node label, $c_i \in \{1, 2, \ldots, Z_v\}$, with a normalized bounding box position, $b_i \in [0, 1]^4$. The bounding box $b_i$ is represented by $(\text{center}_x, \text{center}_y, \text{width}, \text{height})$ and normalized w.r.t. the image canvas size. The edge tensor $E \in \mathbb{R}^{n \times n}$, details the directed edge relationships among the nodes. Each edge entry $e_{i,j}$ corresponds to a discrete relation label, $e_{i,j} \in \{0, 1, \ldots, Z_e\}$, clarifying the connections between nodes. The symbols $Z_v$ and $Z_e$ represent the total numbers of semantic object categories and relation categories of interest, respectively. Notably, $e_{i,j} = 0$ indicates the absence of a relation between nodes $i$ and $j$. The task is to generate such grounded scene graphs from noise.

### 4.1.2 Diffusion Model Basics

**Preliminaries.** Diffusion models (Ho et al., 2020; Song et al., 2021) learn a probabilistic distribution $p_\theta(x)^3$ through matching the score functions of the Gaussian noise perturbed data distribution $\nabla_x \log p_\sigma(x)$ at various noise levels $\sigma \in \{\sigma_i\}_{i=1}^T$. Following Song et al. (2021); Karras et al. (2022), we use the SDE-based diffusion model for training and sampling, which comes with a continuous time $t \in [0, T]$ specified by the following dynamics:

$$dx_+ = f(x, t)dt + g(t)dw, \tag{1}$$

$$dx_- = [f(x, t)dt - g(t)^2 \nabla_x \log p_t(x)]dt + g(t)dw, \tag{2}$$

where Eq. (1) and Eq. (2) denote forward and reverse SDEs, $f(x, t)$ and $g(t)$ are the drift and diffusion coefficients, and $w$ is the standard Wiener process. The SDEs govern how the probabilistic distribution $p_t(x)$ evolves w.r.t. time $t$. Specifically, $p_0(x)$ is the data distribution, from which we observe a set of i.i.d. samples $\mathcal{X} = \{x_i\}_{i=1}^m$. $p_T(x)$ models a tractable prior distribution, *i.e.*, Gaussian, from which we can draw

---

[2]In what follows, we use $I_d$ for identity matrix and $I$ for image data.

[3]We use symbols $x, \tilde{x}$ in this section to introduce preliminaries of diffusion model in general, regardless of the type of data being modeled.

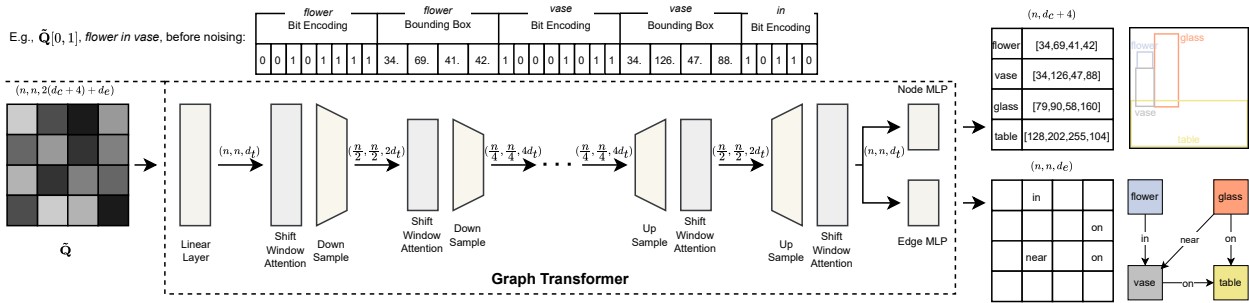

Figure 2: **Illustration of the graph transformer during training.** Object and relation categories are illustrated using binary-bit encoding. Object bounding box positions are represented by $(\text{center}_x, \text{center}_y, \text{width}, \text{height})$.

samples efficiently. In our formulation, we choose linear noise schedule $\sigma(t) = t$, and let $f(\boldsymbol{x}, t) = \boldsymbol{0}$ and $g(t) = \sqrt{2\dot{\sigma}(t)\sigma(t)}$. The SDEs solution yields that $p_t$ in Eq. (2) becomes $p_\sigma(\boldsymbol{x}) = p_t(\boldsymbol{x}) = \frac{1}{m}\sum_{i=1}^{m}\mathcal{N}(\boldsymbol{x}; \boldsymbol{x}_i, \sigma^2\boldsymbol{I_d})$, $\boldsymbol{x}_i \in \mathcal{X}$. Let $p_\mathcal{X}(\boldsymbol{x}) = \frac{1}{m}\sum_{i=1}^{m}\delta(\boldsymbol{x} - \boldsymbol{x}_i)$ be the Dirac delta distribution for $\mathcal{X}$. We can rewrite $p_\sigma$ as $p_\sigma(\tilde{\boldsymbol{x}}) = \int p_\mathcal{X}(\boldsymbol{x})p_\sigma(\tilde{\boldsymbol{x}}|\boldsymbol{x})d\boldsymbol{x}$ with a Gaussian perturbation kernel $p_\sigma(\tilde{\boldsymbol{x}}|\boldsymbol{x}) = \mathcal{N}(\tilde{\boldsymbol{x}}; \boldsymbol{x}, \sigma^2\boldsymbol{I_d})$.

**Training.** We train a neural network to learn the score function of $p_\sigma$ (*i.e.*, $\nabla_{\tilde{\boldsymbol{x}}}\log p_\sigma(\tilde{\boldsymbol{x}})$). Following Vincent (2011); Karras et al. (2022), we reparameterize the score function by denoising function $D(\tilde{\boldsymbol{x}}, \sigma)$ which maps the noise-corrupted data $\tilde{\boldsymbol{x}}$ back to the clean data $\boldsymbol{x}$. They are connected by Tweedie's formula (Efron, 2011), $\nabla_{\tilde{\boldsymbol{x}}}\log p_\sigma(\tilde{\boldsymbol{x}}) = (D(\tilde{\boldsymbol{x}}, \sigma) - \tilde{\boldsymbol{x}})/\sigma^2$. In practice, we train a denoiser $D_\theta(\tilde{\boldsymbol{x}}, \sigma)$ to implicitly capture the score function. Given a specified distribution over the noise level, denoted as $p(\sigma)$, which also corresponds to the distribution of forward time $t$ since $\sigma(t) = t$, the overall training objective can be formulated as follows,

$$\mathbb{E}_{p(\sigma)p_\mathcal{X}(\boldsymbol{x})p_\sigma(\tilde{\boldsymbol{x}}|\boldsymbol{x})}\left[\|D_\theta(\tilde{\boldsymbol{x}}, \sigma) - \boldsymbol{x}\|_2^2\right]. \tag{3}$$

**Sampling.** To draw samples using the learned diffusion model, we discretize the reverse-time SDE in Eq. (2) and conduct numerical integration, which gradually transitions samples from prior distribution $p_T$ to data distribution $p_0$. We choose a set of discrete time steps $\{t_i\}_{i=1}^{T}$, at which the score function is evaluated using the trained model for numerical reverse-SDE solution. We employ a second-order solver based on Heun's method (Süli & Mayers, 2003; Karras et al., 2022).

### 4.1.3 DiffuseSG for Grounded Scene Graph Generation

We model the grounded scene graph distribution $p(\boldsymbol{S})$ with a continuous state diffusion model. Our model captures the distribution of grounded scene graph topology along with node and edge attributes simultaneously.

**Denoising Objective.** During training, we draw noisy grounded scene graph samples from the perturbed distribution $p_\sigma(\tilde{\boldsymbol{S}}) = \int p_\mathcal{S}(\boldsymbol{S})p_\sigma(\tilde{\boldsymbol{S}}|\boldsymbol{S})d\boldsymbol{S}$ and train a denoiser $D_\theta(\tilde{\boldsymbol{S}}, \sigma)$ to output the associated noise-free samples $\boldsymbol{S}$. We relax node and edge label distributions to continuous space, enabling SDE-based diffusion modeling with smooth Gaussian noise on all attributes. Various methods for encoding discrete labels will be introduced in Section "Encoding Discrete Data" below. Specifically, $\tilde{\boldsymbol{S}} = (\tilde{\boldsymbol{V}}, \tilde{\boldsymbol{E}})$ denotes the noisy grounded scene graph and $p_\mathcal{S}(\boldsymbol{S}) = \frac{1}{m}\sum_{i=1}^{m}\delta(\boldsymbol{S} - \boldsymbol{S}_i)$ is the Dirac delta distribution based on training data $\{\boldsymbol{S}_i\}_{i=1}^{m}$. We implement the grounded scene graph Gaussian perturbation kernel $p_\sigma(\tilde{\boldsymbol{S}}|\boldsymbol{S})$ by independently injecting noise to node and edge attributes, *i.e.*, $p_\sigma(\tilde{\boldsymbol{S}}) = \int p_\mathcal{S}(\boldsymbol{S})p_\sigma(\tilde{\boldsymbol{V}}|\boldsymbol{V})p_\sigma(\tilde{\boldsymbol{E}}|\boldsymbol{E})d\boldsymbol{S}$. The decomposed kernels are both simple Gaussians: $p_\sigma(\tilde{\boldsymbol{V}}|\boldsymbol{V}) = \mathcal{N}(\tilde{\boldsymbol{V}}; \boldsymbol{V}, \sigma^2\boldsymbol{I_d})$, $p_\sigma(\tilde{\boldsymbol{E}}|\boldsymbol{E}) = \mathcal{N}(\tilde{\boldsymbol{E}}; \boldsymbol{E}, \sigma^2\boldsymbol{I_d})$. Further, we design a denoising network $D_\theta$ with two prediction heads $D_\theta^V$ and $D_\theta^E$ dedicated to node and edge attributes respectively (detailed in Section "Network Design" below). Our grounded scene graph denoising loss now becomes:

$$\mathcal{L}_d = \mathbb{E}_{p(\sigma)p_\mathcal{S}(\boldsymbol{S})p_\sigma(\tilde{\boldsymbol{S}}|\boldsymbol{S})}[\|D_\theta^V(\tilde{\boldsymbol{S}}, \sigma) - \boldsymbol{V}\|_2^2 + \|D_\theta^E(\tilde{\boldsymbol{S}}, \sigma) - \boldsymbol{E}\|_2^2]. \tag{4}$$

---

**Algorithm 1** DiffuseSG Training Process.

---

**Require:** node denoiser $D_\theta^V$, edge denoiser $D_\theta^E$, diffusion time distribution $p(\sigma)$, dataset $\mathcal{X}$.

1: **repeat**
2:      **sample** $\boldsymbol{S} \sim \mathcal{X}$           ▷ Draw one grounded scene graph sample
3:      **sample** $\sigma \sim p(\sigma)$           ▷ Draw a diffusion timestamp
4:      $(\boldsymbol{V}, \boldsymbol{E}) \leftarrow \boldsymbol{S}$           ▷ Obtain the node and edge tensors
5:      $\boldsymbol{B} \leftarrow \boldsymbol{V}$           ▷ Obtain the bounding box positions
6:      $\tilde{\boldsymbol{V}} \leftarrow \boldsymbol{V} + \sigma\boldsymbol{\epsilon}_1$ with $\boldsymbol{\epsilon}_1 \sim \mathcal{N}(\mathbf{0}, \boldsymbol{I})$           ▷ Add noise to the node tensor
7:      $\tilde{\boldsymbol{E}} \leftarrow \boldsymbol{E} + \sigma\boldsymbol{\epsilon}_2$ with $\boldsymbol{\epsilon}_2 \sim \mathcal{N}(\mathbf{0}, \boldsymbol{I})$           ▷ Add noise to the edge tensor
8:      $\tilde{\boldsymbol{S}} \leftarrow (\tilde{\boldsymbol{V}}, \tilde{\boldsymbol{E}})$
9:      $\hat{\boldsymbol{V}} = D_\theta^V(\tilde{\boldsymbol{S}}, \sigma)$           ▷ Denoise the node tensor
10:      $\hat{\boldsymbol{E}} = D_\theta^E(\tilde{\boldsymbol{S}}, \sigma)$           ▷ Denoise the edge tensor
11:      $\mathcal{L}_d = \|\hat{\boldsymbol{V}} - \boldsymbol{V}\|_2^2 + \|\hat{\boldsymbol{E}} - \boldsymbol{E}\|_2^2$           ▷ Eq. (4)
12:      $\hat{\boldsymbol{B}} \leftarrow \hat{\boldsymbol{V}}$           ▷ Obtain the denoised bounding box positions
13:      $\mathcal{L}_{iou} = 1 - \frac{1}{n} \sum_{i=1}^n \text{GIoU}(\hat{\boldsymbol{B}}_i, \boldsymbol{B}_i)$           ▷ Eq. (5)
14:      **update** $\theta$ via $-\nabla_\theta(\mathcal{L}_d + \lambda\mathcal{L}_{iou})$           ▷ Optimization step
15: **until** converged

---

**Network Design.** To effectively capture the complex distribution of grounded scene graphs, we develop a transformer architecture, named *graph transformer*, as the denoiser $D_\theta$. Given a noisy grounded scene graph $\tilde{\boldsymbol{S}} = (\tilde{\boldsymbol{V}}, \tilde{\boldsymbol{E}})$ as input, we construct triplet representations (*i.e.*, generalized edge representations) by concatenating the subject node, object node, and relation information $\tilde{\boldsymbol{Q}}[i,j] = [\tilde{\boldsymbol{v}}_i, \tilde{\boldsymbol{v}}_j, \tilde{\boldsymbol{e}}_{i,j}], \forall i, j \in [n]$, where $n$ is the number of nodes in the grounded scene graph. The denoising task is essentially node and edge regression in continuous space, trained with stochasticity. For expressive graph representation learning in this context, we consider message passing among all $O(n^2)$ triplets as suggested in Morris et al. (2019; 2021). Here, each triplet becomes a unit of message passing. However, a naive triplet-to-triplet message passing implementation is space-consuming ($O(n^4)$ messages). Inspired by Liu et al. (2021), we employ an approximate triplet-to-triplet message passing using shifted-window attention layers with a window size $M$, which reduces the space complexity to $O(n^2 M^2)$. When window-partitioning is repeated adequately, *e.g.*, at least $O(n/M)$ times, all triplet-to-triplet interactions can be effectively approximated. We tokenize the noisy triplets using a linear layer on each entry in $\tilde{\boldsymbol{Q}}$, resulting in $n \times n$ triplet tokens of dimension $d_t$, represented as $\tilde{\boldsymbol{Q}}_d \in \mathbb{R}^{n \times n \times d_t}$. Our graph transformer then employs repeated shifted-window attention and downsampling/upsampling layers to update the dense triplet-token representations, for predicting the noise-free node and edge attributes. To generate node and edge attribute predictions of distinct shapes, we employ two MLPs as readout layers for node and edge respectively. The node denoiser $D_\theta^V$ and the edge denoiser $D_\theta^E$ share identical intermediate feature maps and have the same parameters up to the final readout layers. The network architecture is further detailed in Appendix A.2.

**Encoding Discrete Data.** To find an appropriate representation for the categorical labels, denoting the node label representation as $c_i' \in \mathbb{R}^{d_c}$ and the edge label representation as $e_{i,j}' \in \mathbb{R}^{d_e}$, we explore three distinct encoding methods. (1) Scalar: both node and edge labels are expressed as scalar values, with $d_c = d_e = 1$. (2) Binary-Bit Encoding: the discrete type indices of node and edge labels are converted into their binary format, represented as a sequence of 0s and 1s. Here, $d_c = \lceil \log_2(Z_v) \rceil$ and $d_e = \lceil \log_2(Z_e + 1) \rceil$. (3) One-Hot Encoding: the scalar labels are transformed into their one-hot representations, resulting in $d_c = Z_v$ and $d_e = Z_e + 1$. The impact of these different encoding methods is further analyzed and compared in Sec. 5.2.3. During training, after encoding the node and edge labels, along with the bounding box positions, we inject noise and form the noisy $\tilde{\boldsymbol{Q}}$. Fig. 2 illustrates the pipeline of our graph transformer during training. During sampling, we start with a Gaussian noised $\tilde{\boldsymbol{Q}}$, where the node number can either be specified or drawn from some given distribution. After the denoising process, we discretize the continuous-valued representations of node and edge types, *e.g.*, through thresholding for the binary-bit encoding. Detailed explanations of these categorical encodings and discretizations during sampling are in Appendix A.1. Note that the bounding box positions $\boldsymbol{b}_i$ are naturally continuous and there is no need for discretization while sampling.

**Additional Bounding Box IoU Loss.** To enhance bounding box generation quality, we integrate an intersection-over-union (IoU)-based loss, $\mathcal{L}_{iou}$, into the denoising objective. This IoU loss aims to align the denoised bounding boxes $\hat{\boldsymbol{B}} \in \mathbb{R}^{n \times 4}$ (a partial output of $D_\theta^V$) closely with the ground-truth $\boldsymbol{B} \in \mathbb{R}^{n \times 4}$. The IoU loss is formulated as:

$$\mathcal{L}_{iou} = 1 - \frac{1}{n} \sum_{i=1}^{n} \text{GIoU}(\hat{\boldsymbol{B}}_i, \boldsymbol{B}_i), \tag{5}$$

where $n$ is the number of objects in the grounded scene graph, and GIoU is the generalized IoU, proposed in Rezatofighi et al. (2019), of the corresponding boxes. The final training loss then becomes:

$$\mathcal{L} = \mathcal{L}_d + \lambda \mathcal{L}_{iou}, \tag{6}$$

where $\lambda$ is a hyperparameter that adjusts the balance between these two loss components. Note, $\mathcal{L}_d$ is given in Eq. (4). We use $\lambda = 1$ in our experiments. Algorithm 1 shows the training process of DiffuseSG. Note, the node and edge denoisers ($D_\theta^V$ and $D_\theta^E$) share identical intermediate feature maps and have the same parameters up to the final prediction heads, and the bounding box positions are part of the node tensor. The diffusion modeling details are in Appendix A.3.

**Summary of Technical Contributions.** Though the shifted window attention is adopted from Liu et al. (2021), and the U-Net architecture is similar to other diffusion models like the ones in Song et al. (2021); Karras et al. (2022). Our novelties lie in: (1) the tensor representation of the grounded scene graph for DiffuseSG (a continuous diffusion model), (2) the separate read out layers (MLP prediction heads) for object and relation generation, (3) an additional bounding box IoU loss for training, and (4) exploration among different encoding mechanisms for object and relation categories.

## 4.2 Conditional Image Generation

We use two diffusion-based conditional image generators to model the conditional image distribution $p(\boldsymbol{I}|\boldsymbol{S})$ given generated grounded scene graphs: one existing layout-to-image generator named *LayoutDiffusion* (Zheng et al., 2023b), and one relation-aware image generator which is built upon ControlNet (Zhang et al., 2023b) by ourselves, termed *Relation-ControlNet*.

### 4.2.1 LayoutDiffusion

LayoutDiffusion (Zheng et al., 2023b), is a diffusion-based layout-to-image generation model. It uses a U-Net architecture for the denoising process, with the layout condition enforced on the hidden features of the U-Net. It first employs a transformer-based layout fusion module to capture the information in the given layout, and then utilizes a cross-attention mechanism to fuse the image features and the layout representations inside the denoising U-Net. The whole diffusion process is applied on the image pixel space. Following Ho et al. (2020); Ho & Salimans (2022), LayoutDiffusion utilizes a standard mean-squared error loss to train the diffusion model and the classifier-free guidance technique to support the layout condition.

LayoutDiffusion is trained and evaluated on the Visual Genome (Krishna et al., 2017) and COCO-Stuff (Caesar et al., 2018) datasets, which is aligned with our dataset settings as described in Sec. 5.1. Given the superior performance of LayoutDiffusion on these two datasets, we decide to adopt it as one of our conditional image generator candidates. Specifically, we take the model checkpoints provided by the authors[4] which generate images of resolution $256 \times 256$.

### 4.2.2 Relation-Aware Layout-to-Image Generation (Relation-ControlNet)

As discussed in Sec. 3, the existing conditional image generators are either conditioned only on object labels and their locations (layout-to-image models), or object labels and relation labels but no object locations (scene graph-to-image models). To fully utilize the data information generated by our DiffuseSG, we build a relation-aware layout-to-image generator, which generates images conditioned on object labels, locations and relation labels. We call the model, which is based on ControlNet (Zhang et al., 2023b), Relation-ControlNet.

---

[4]https://github.com/ZGCTroy/LayoutDiffusion

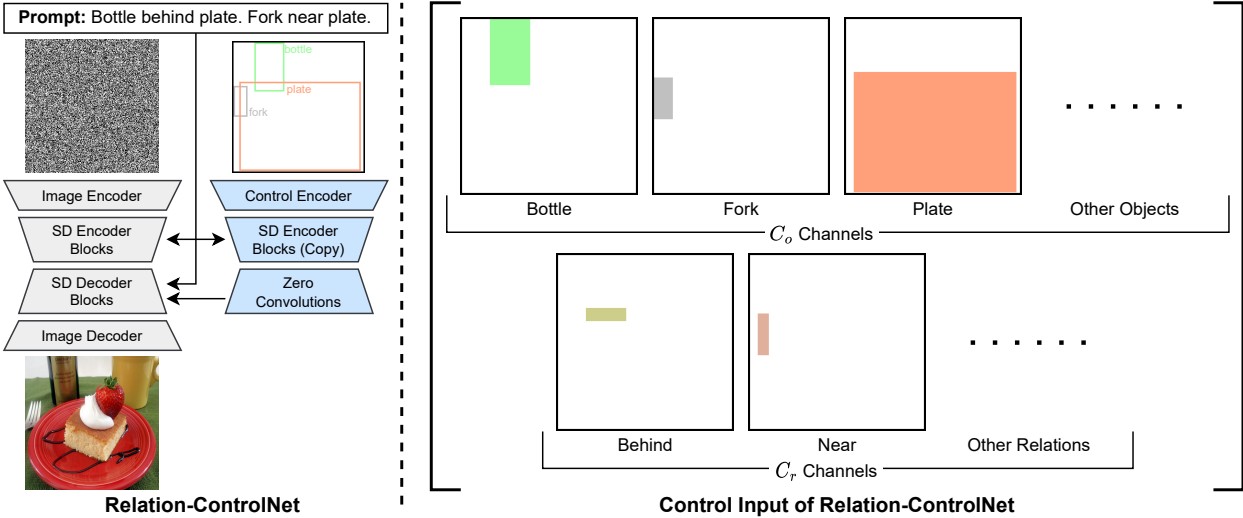

Figure 3: **Relation-ControlNet Illustration.** The control input is to provide spatial information for both objects and relations. Each channel in the control input has size $(256, 256)$, where the colored areas having values of 1 and the white areas are filled with values of 0. In the figure, SD stands for Stable Diffusion.

**ControlNet.** ControlNet (Zhang et al., 2023b) is a neural network architecture which can add spatial conditioning controls to large pre-trained text-to-image diffusion models. When it is instantiated on the pre-trained Stable Diffusion (Rombach et al., 2022) model, the encoder part of the U-Net is first cloned. The image feature is inputted to both the original U-Net encoder and the cloned counterpart, while the cloned network blocks receive an additional control input. The output of the cloned U-Net encoder is then injected into the original U-Net decoder via *zero convolution layers*. The control input has the same spatial dimension as the input/output image. Since the U-Net in Stable Diffusion works on the latent image features, the control input is also transformed by convolution layers to match the input dimension of the diffusion U-Net. The original training loss in Stable Diffusion is used. During training, only the weights of the cloned U-Net encoder and the newly introduced convolution layers get updated. We utilize ControlNet with Stable Diffusion V1.5 to build our relation-aware layout-to-image generation model. Specifically, we tailor the control input and the text prompt of ControlNet to encode the object label, object location, relation label, and relation location information. We train the ControlNet to produce images of resolution $256 \times 256$.

**Control Input.** The control input to the ControlNet is to specify the spacial information. We utilize this to provide the spacial information of both objects and relations given a grounded scene graph. Specifically, assuming there are $C_o$ object categories and $C_r$ relation classes given a dataset, we construct a control input of size $(256, 256, C_o + C_r)$ to encode the object and relation locations, where we use one channel to represent one specific object/relation category. That is, the channel $C_i$ is to provide the location information of object/relation class $C_i$, and all objects or relations having the same class are encoded in the same channel. The first $C_o$ channels are for object classes while the last $C_r$ channels are for relation categories. The relation location is determined via the "between" operation given the subject and object bounding boxes as defined in Hoe et al. (2024). The bounding box positions are mapped from $[0, 1]$ in continuous space to $[0, 1, ..., 255]$ discrete grids to form the control input.

**Text Prompt.** Given the specific version of Stable Diffusion (V1.5) used, the text prompt encoder can only encode 77 tokens. Therefore, we let the text prompt only include the relation information. We define a relation sentence as a concatenation of the subject label, the relation label, and the object label for a relation triplet, ending with a period, *e.g.*, "cat under chair.". Given the limited capability of the text encoder, we decide to only encode the unique relation sentences given a grounded scene graph. Same as in Zhang et al. (2023b), during training, the text prompt is replaced with an empty string at a probability of 50%.

The Relation-ControlNet is illustrated in Fig. 3, with emphasis on the control input. The training details can be found in Appendix A.4.

## 5 Experiments

We conduct all experiments on the Visual Genome (Krishna et al., 2017) and COCO-Stuff (Caesar et al., 2018) datasets. Our experiments mainly focus on the grounded scene graph generation part since this is the main contribution of this work.

### 5.1 Datasets

**Visual Genome (VG).** We use the same pre-processing procedure and train/val splits as previous grounded scene graph detection works (Xu et al., 2017; Zellers et al., 2018; Tang et al., 2020; Li et al., 2022a), but only the grounded scene graph annotations are used. This pre-processed dataset contains $57,723$ training and $5,000$ validation grounded scene graphs with 150 object and 50 relation categories. For each grounded scene graph, we ensure that there will be only one edge label if there exists a directed edge (relation) between two nodes (objects). Each grounded scene graph has node numbers between 2 and 62, with an average of 5.15 relations per instance.

**COCO-Stuff.** COCO-Stuff contains 171 object types (including 80 thing and 91 stuff categories). It comes with object label and bounding box annotations, but no relation labels. Following Johnson et al. (2018), we manually assign a relation label between two bounding boxes based on their relative positions, with a relation set of 6 labels: `left of`, `right of`, `above`, `below`, `inside`, and `surrounding`. Same as in Kong et al. (2022), we remove small bounding boxes ($\leq 2\%$ image area) and instances tagged as "iscrowd", resulting in $118,262$ training and $4,999$ validation grounded scene graphs. Each grounded scene graph is a fully-connected graph without self-loop, with node numbers between 1 and 33.

### 5.2 Grounded Scene Graph Generation Experiments

#### 5.2.1 Evaluation Metrics

We use maximum mean discrepancy (MMD), triplet label total variation difference (Triplet TV), and our proposed novel object detection-based F1 scores to measure the model performance on the grounded scene graph generation task.

**MMD.** Inspired by the relevant graph generation literature (You et al., 2018; Liao et al., 2019), we use MMDs to measure the similarities between the generated grounded scene graphs and the ground-truth ones on the node degree, node label, and edge label distributions respectively. Empirically, let $\{x_i\}_{i=1}^n$ be the generated samples and $\{y_j\}_{j=1}^m$ be the ground-truth ones. The MMD value is calculated as $\frac{1}{n^2}\sum_{i=1}^n\sum_{j=1}^n k(x_i, x_j) + \frac{1}{m^2}\sum_{i=1}^m\sum_{j=1}^m k(y_i, y_j) - \frac{2}{nm}\sum_{i=1}^n\sum_{j=1}^m k(x_i, y_j)$, where $n$ and $m$ are the numbers of generated samples and ground-truth ones respectively, and we use the Gaussian kernel as the kernel function $k(\cdot, \cdot)$. The lower MMD value means the closer to the ground-truth distribution.

**Triplet TV.** As the labels of the <`subject`, `relation`, `object`> triplets lie in a very high dimension (the cross product of potential subject, relation, and object types), it is computationally infeasible to calculate the triplet label MMD. As a compromise, we use the total variation difference (TV) to measure the marginal distribution difference between the generated triplet labels and the ground-truth ones. Specifically, assuming that the generated empirical distribution is $\hat{p}$ with the ground-truth being $\hat{q}$, where $\hat{p}$ and $\hat{q}$ are vectors sized as the number of unique triplets combining all generated and ground-truth triplets. The TV is calculated as $\frac{1}{2}|\hat{p} - \hat{q}|$. Note, if a triplet does not exist in either $\hat{p}$ or $\hat{q}$, then its relevant entry in $\hat{p}$ or $\hat{q}$ is 0.

**Detection-based F1 Scores.** We propose a set of novel object detection-based F1 scores to evaluate the generated bounding box layout quality (including both the location and node label). Specifically, for a generated layout, we calculate a F1 score between this generated layout and every ground-truth layout, and take the maximum one as the final score. Assume there are $N$ node categories. Given a pair of generated and ground-truth layouts, the F1 score is calculated as $\text{F1} = \sum_{c \in N}(w_c \cdot \text{F1}_c)$, where $\text{F1}_c$ is F1 score for

Table 2: **Different attributes** that can be generated from different methods.

| Method | Generation Type | Attributes Generated | | |
|---|---|---|---|---|
| | | Object Label | Object Location | Relation Label |
| SceneGraphGen | Scene Graph | ✓ | | ✓ |
| VarScene | Scene Graph | ✓ | | ✓ |
| D3PM | Scene Graph | ✓ | | ✓ |
| BLT | Layout | ✓ | ✓ | |
| LayoutDM | Layout | ✓ | ✓ | |
| DiGress | Grounded Scene Graph | ✓ | ✓ | ✓ |
| DiffuseSG | Grounded Scene Graph | ✓ | ✓ | ✓ |

a node category $c$ and $w_c$ is its weighting coefficient. Calculating $F1_c$ needs to decide whether two bounding boxes match or not. We use 10 different bounding box IoU thresholds ranging from 0.05 to 0.5 with a step size of 0.05 to decide the bounding box match. That is, $F1_c = \frac{1}{10} \sum_{\text{iou} \in [0.05:0.05:0.5]} F1(\text{iou}|c)$, where $F1(\text{iou}|c)$ means a F1 score between two layouts given a specific node category $c$ and a IoU threshold iou. We calculate 4 different types of F1 scores: (1) F1-Vanilla (F1-V), where $w_c$ is set to $\frac{1}{|N|}$ for every node category; (2) F1-Area (F1-A), where $w_c$ is set to $\frac{\text{Area}(c)}{\sum_{c \in N} \text{Area}(c)}$ and $\text{Area}(c)$ is the average bounding box area in the validation set for the node category $c$; (3) F1-Frequency (F1-F), where $w_c$ is set to $\frac{\text{Freq}(c)}{\sum_{c \in N} \text{Freq}(c)}$ and $\text{Freq}(c)$ is the frequency of the node category $c$ in the validation set; (4) F1-BBox Only (F1-BO), where the F1 calculation is purely based on the bounding box locations, that is, we treat all bounding boxes as having a single node category ($|N| = 1$ and $w_c = 1$). The motivation of having F1-Area and F1-Frequency is that we want some metrics to be slightly biased to those salient objects (appearing either in a large size in general or more frequently). The higher the F1 scores, the better.

### 5.2.2 Baselines

We consider the following six baselines to compare to our DiffuseSG model.

(1) **SceneGraphGen** (Garg et al., 2021) is a scene graph generative model based on recurrent networks. It is an autoregressive model where one object label or one relation label is generated at a time. This model is not capable of producing object bounding boxes, and we can not specify the number of objects in a scene graph to be generated during inference.

(2) **VarScene** (Verma et al., 2022) is a variational autoencoder-based generative model for scene graphs. Similar to SceneGraphGen, it generates scene graphs without bounding boxes and can not accept number of objects as parameter during inference. Using the released code[5], we successfully replicate the results on the VG dataset but encounter issues with COCO-Stuff, as its symmetric modeling of edges between nodes prevents VarScene from generating directed graphs. Consequently, we report directed graph results on VG but undirected graph results on COCO-Stuff.

(3) **D3PM** (Austin et al., 2021) is a discrete denoising diffusion probabilistic framework designed for discrete data generation. We adopt its image generation model for scene graph generation; details are in Appendix B. This model only generates node and edge labels.

(4) **BLT** (Kong et al., 2022) is a transformer-based layout generation model where only object labels and bounding boxes are generated. The transformer is non-autoregressive, where all the attributes of the layout are generated at the same time as discrete tokens. Bounding box locations are quantized into integers.

(5) **LayoutDM** (Inoue et al., 2023) is a discrete state-space diffusion model for layout generation. Similar to BLT, this model is also not able to model relation labels and the bounding box locations are discretized.

---

[5]https://github.com/structlearning/varscene/tree/main

Table 3: **Grounded scene graph generation results** on the Visual Genome and COCO-Stuff validation sets. In each column, the best value is **bolded**. N-MMD, D-MMD, and E-MMD are the MMD values calculated based on node label distribution, node degree distribution, and edge label distribution respectively. T-TV (val) / (train) is the Triplet TV calculated against validation / training triplet statistics. The training set has a larger set of triplets than the validation, giving a more comprehensive evaluation.

| Visual Genome (VG) | | | | | | | | | |
|---|---|---|---|---|---|---|---|---|---|
| Method | N-MMD↓ | F1-V↑ | F1-A↑ | F1-F↑ | F1-BO↑ | D-MMD↓ | E-MMD↓ | T-TV (val)↓ | T-TV (train)↓ |
| LayoutDM | 9.44e-3 | 0.161 | 0.291 | 0.368 | **0.766** | - | - | - | - |
| SceneGraphGen | **8.77e-3** | - | - | - | - | 3.79e-2 | **2.29e-2** | 0.987 | 0.979 |
| VarScene | 2.58e-2 | - | - | - | - | 1.04e-2 | 3.91e-2 | 0.988 | 0.981 |
| DiffuseSG* | 9.52e-3 | **0.188** | **0.331** | **0.369** | 0.749 | **6.35e-3** | 3.25e-2 | **0.735** | **0.566** |
| BLT | 2.70e-2 | 0.181 | 0.300 | **0.376** | 0.708 | - | - | - | - |
| D3PM | 7.69e-3 | - | - | - | - | 3.07e-2 | 2.00e-2 | 0.816 | 0.772 |
| DiGress | 7.94e-3 | 0.157 | 0.263 | 0.282 | 0.732 | 8.89e-3 | **8.02e-3** | 0.718 | 0.706 |
| DiffuseSG | **6.64e-3** | **0.184** | **0.308** | 0.292 | **0.747** | **5.26e-3** | 3.46e-2 | **0.702** | **0.685** |

| COCO-Stuff | | | | | | | | | |
|---|---|---|---|---|---|---|---|---|---|
| Method | N-MMD↓ | F1-V↑ | F1-A↑ | F1-F↑ | F1-BO↑ | D-MMD↓ | E-MMD↓ | T-TV (val)↓ | T-TV (train)↓ |
| LayoutDM | **3.40e-4** | 0.274 | 0.330 | 0.508 | **0.824** | - | - | - | - |
| SceneGraphGen | 3.79e-4 | - | - | - | - | 2.59e-3 | 7.24e-4 | 0.904 | 0.895 |
| VarScene | 2.60e-2 | - | - | - | - | 3.07e-1 | 9.32e-2 | 0.949 | 0.949 |
| DiffuseSG* | 1.22e-3 | **0.439** | **0.500** | **0.639** | 0.822 | **1.44e-4** | **1.59e-4** | **0.229** | **0.287** |
| BLT | 1.09e-1 | 0.322 | 0.389 | 0.526 | 0.807 | - | - | - | - |
| D3PM | **4.92e-4** | - | - | - | - | 0 | 1.29e-4 | 0.341 | 0.305 |
| DiGress | 1.06e-3 | 0.342 | 0.387 | 0.570 | 0.782 | 0 | 4.44e-3 | 0.515 | 0.398 |
| DiffuseSG | 5.53e-4 | **0.421** | **0.485** | **0.637** | **0.830** | 0 | **7.25e-5** | **0.270** | **0.219** |

(6) **DiGress** (Vignac et al., 2022) is a discrete denoising diffusion model for generating graphs with categorical node and edge labels. We add an additional input of discrete bounding box representation, same as the one used in LayoutDM, to incorporate bounding box generation.

Tab. 2 shows the different attributes that can be generated among all the baselines and DiffuseSG. Among the six baselines, SceneGraphGen, VarScene, and LayoutDM can not deal with specification on the number of objects for the scene graphs or layouts to be generated during sampling, while the other three can. Comparing to those diffusion-based baselines (D3PM, LayoutDM, and DiGress), DiffuseSG performs the diffusion process in the continuous space. For all the baselines, we train them from scratch using the authors' released code, with slight adaptation to the datasets.

### 5.2.3 Grounded Scene Graph Generation Results

Our DiffuseSG is trained with Eq. (6) and uses the binary-bit input representation. To compare with SceneGraphGen, VarScene, and LayoutDM, while sampling with DiffuseSG, we do not fix the object numbers but draw the number of objects from its empirical distribution on the validation set. This line of results is denoted as DiffuseSG*. Since the other three models (D3PM, BLT, and DiGress) can specify the object numbers during inference, when comparing with these three models, the ground-truth object numbers are used for grounded scene graph generation. For all our models and baselines, we consistently randomly sample a fixed set of $1,000$ training samples to do model selection.

Tab. 3 shows the quantitative results. For each model, we generate 3 sets of layout, scene graph, or grounded scene graph samples, where the size of one set samples is equal to the number of instances in the respective validation set. Reported results are the averaged results over the 3 sample sets. From the table, we can see that our DiffuseSG* and DiffuseSG achieves the best results on most evaluation metrics. Comparing DiffuseSG* with LayoutDM, SceneGraphGen, and VarScene, DiffuseSG* is better than these baselines on F1-V, F1-A, and F1-F scores (describing both object labels and bounding box locations), D-MMD (measuring the graph connectivity), and Triplet-TV (capturing the co-occurrence between objects and their relevant

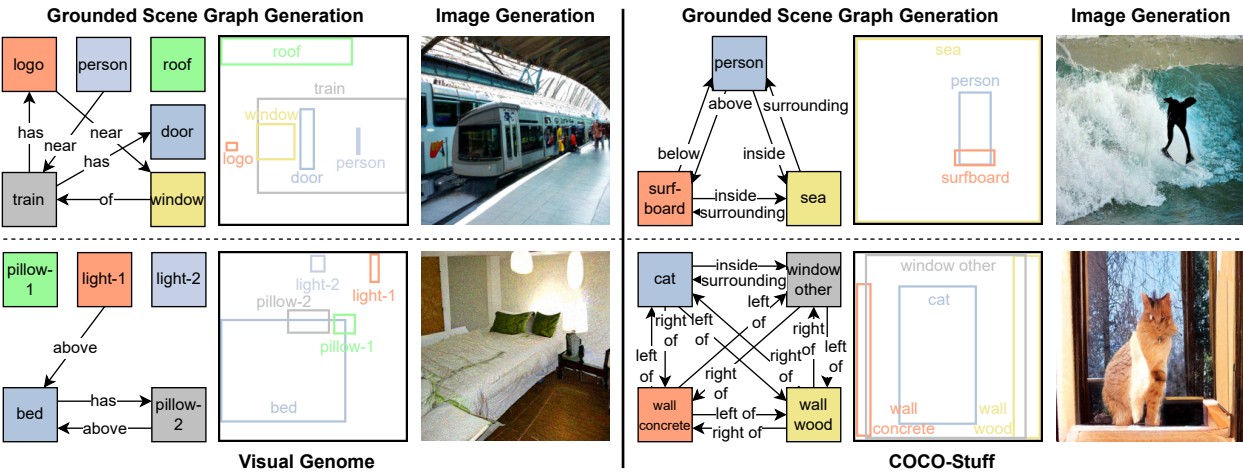

Figure 4: **Grounded scene graph - image pair generation qualitative results.** Grounded scene graphs are generated by DiffuseSG and the corresponding $256 \times 256$ images are produced by LayoutDiffusion (Zheng et al., 2023b).

Table 4: **Ablations** regarding the different categorical input representations and the IoU loss on both the Visual Genome and COCO-Stuff validation sets. The model without the IoU loss is denoted as DiffuseSG$^-$. The evaluation metrics are the same as the ones used in Tab. 3.

| Visual Genome (VG) | | | | | | | | | |
|---|---|---|---|---|---|---|---|---|---|
| Method | N-MMD↓ | F1-V↑ | F1-A↑ | F1-F↑ | F1-BO↑ | D-MMD↓ | E-MMD↓ | T-TV (val)↓ | T-TV (train)↓ |
| DiffuseSG (bit) | 6.64e-3 | **0.184** | **0.308** | **0.292** | 0.747 | 5.26e-3 | 3.46e-2 | **0.702** | **0.685** |
| DiffuseSG (scalar) | **4.91e-3** | 0.109 | 0.210 | 0.262 | 0.742 | **4.91e-3** | 7.61e-2 | 0.982 | 0.824 |
| DiffuseSG (one-hot) | 2.94e-2 | 0.108 | 0.196 | 0.260 | **0.802** | 1.84e-2 | **2.96e-2** | 0.978 | 0.825 |
| DiffuseSG$^-$ (bit) | 6.57e-3 | **0.173** | **0.285** | **0.283** | **0.736** | 7.85e-3 | **3.40e-2** | **0.709** | **0.692** |
| DiffuseSG$^-$ (scalar) | 8.65e-3 | 0.168 | 0.267 | 0.276 | 0.712 | **7.69e-3** | 4.77e-2 | 0.729 | 0.713 |
| DiffuseSG$^-$ (one-hot) | **3.05e-3** | 0.142 | 0.249 | 0.253 | 0.689 | 8.94e-3 | 5.77e-2 | 0.795 | 0.751 |
| COCO-Stuff | | | | | | | | | |
| Method | N-MMD↓ | F1-V↑ | F1-A↑ | F1-F↑ | F1-BO↑ | D-MMD↓ | E-MMD↓ | T-TV (val)↓ | T-TV (train)↓ |
| DiffuseSG (bit) | **5.53e-4** | **0.421** | **0.485** | **0.637** | **0.830** | 0 | 7.25e-5 | **0.270** | **0.219** |
| DiffuseSG (scalar) | 1.24e-2 | 0.172 | 0.222 | 0.368 | 0.822 | 0 | **1.41e-5** | 0.895 | 0.915 |
| DiffuseSG (one-hot) | 1.86e-2 | 0.161 | 0.206 | 0.317 | 0.824 | 0 | 2.89e-3 | 0.807 | 0.793 |
| DiffuseSG$^-$ (bit) | **5.63e-4** | **0.422** | **0.481** | **0.634** | **0.821** | 0 | **7.94e-5** | **0.272** | **0.225** |
| DiffuseSG$^-$ (scalar) | 8.72e-4 | 0.380 | 0.432 | 0.605 | 0.788 | 0 | 4.65e-4 | 0.312 | 0.282 |
| DiffuseSG$^-$ (one-hot) | 2.35e-3 | 0.365 | 0.372 | 0.553 | 0.762 | 0 | 1.82e-3 | 0.439 | 0.332 |

edge labels). Notably, DiffuseSG* is better than the second best model SceneGraphGen by 83.25% (VG) and 94.44% (COCO-Stuff) regarding D-MMD, and by 25.53% (VG) and 74.67% with respect to T-TV (val). DiffuseSG* is only worse than SceneGraphGen on N-MMD (VG) and E-MMD (VG), and LayoutDM on N-MMD (COCO-Stuff) and F1-BO (VG and COCO-Stuff); but the differences are marginal. Comparing DiffuseSG with BLT, D3PM, and DiGress, DiffuseSG is better than these baselines on all F1 scores (except F1-F on VG), and Triplet-TV. Remarkably, on COCO-Stuff, DiffuseSG is 23.10%, 25.32%, and 11.75% better than DiGress with regard to F1-V, F1-A, and F1-F respectively. DiffuseSG is only worse than D3PM on N-MMD (COCO-Stuff), BLT on F1-F (VG), and DiGress on E-MMD (VG); but all the gaps are small.

**Qualitative Results.** Some qualitative results of DiffuseSG are shown in Fig. 4 (more in Appendix D.1). As can be seen, the generated grounded scene graphs are reasonable. On VG, DiffuseSG learns the sparsity of the semantic edges. While on COCO-Stuff, DiffuseSG captures the fully-connected graph pattern.

Table 5: **Grounded scene graph completion results** on VG validation set.

| Method | Single Node Label Completion | | | | | Single Edge Label Completion | | | | |
|---|---|---|---|---|---|---|---|---|---|---|
| | HR@1 | HR@10 | HR@50 | HR@100 | mA | HR@1 | HR@10 | HR@25 | HR@50 | mA |
| DiGress | 12.4 | 21.2 | 65.3 | 91.2 | 20.7 | 9.8 | 30.9 | 41.4 | 63.2 | 15.8 |
| DiffuseSG | **13.9** | **25.7** | **73.2** | **94.5** | **23.6** | **10.1** | **35.7** | **46.2** | **65.3** | **19.4** |

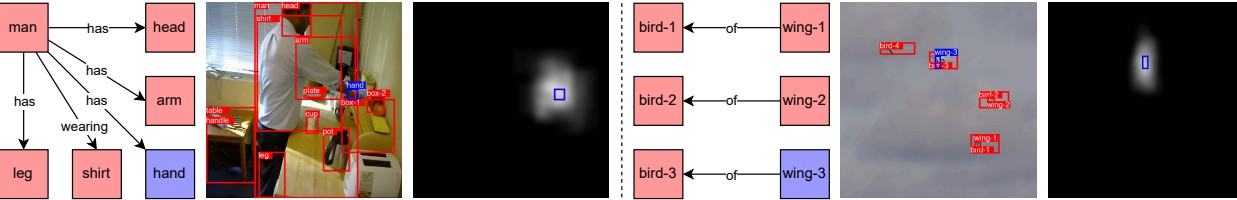

Figure 5: **Single bounding box completion.** The left figure shows the input grounded scene graph, where only the edges and corresponding node labels are shown. The blue node's bounding box has been masked out. The middle figure shows the untouched (input) bounding boxes with labels in red, the one masked out in blue, along with the corresponding ground-truth image. The right figure shows our generated bounding box heatmap in white along with the target ground-truth bounding box (to be completed) in blue. The heatmap is obtained via generating the bounding box 100 times; the whiter the area, the more overlap at the location.

**Ablations.** We conduct ablations with different categorical input representations (binary-bit, scalar, and one-hot), both with and without the IoU loss (Eq. (5)). Tab. 4 shows the ablation results, where the model without the IoU loss is denoted as DiffuseSG⁻. Comparing among the three different categorical input encoding methods on both DiffuseSG and DiffuseSG⁻, the binary-bit representation works the best on both Visual Genome and COCO-Stuff datasets. Comparing DiffuseSG (bit) with DiffuseSG⁻ (bit) on the F1 scores, we can see the effectiveness of our proposed IoU loss. The IoU loss seems not working well while using scalar and one-hot encodings. We believe that with these two types of representations the hyperparameters need to be further tuned. Current set of hyperparameters is tuned using the binary-bit representation. Overall, using the binary-bit representation with the IoU loss gives us the best performance on both datasets.

### 5.2.4 Grounded Scene Graph Completion

Our DiffuseSG is versatile; besides doing the pure grounded scene graph generation, it can also achieve a variety of grounded scene graph completion tasks. We mainly follow Lugmayr et al. (2022) for all the completions. All completion tasks are conducted on VG.

**Single Node Label Completion.** In this setting, we randomly masked out one node label per grounded scene graph in the validation set. For each grounded scene graph, we keep all the bounding box locations and edge labels, and let the model complete one node label given the remaining ones. The node whose label is masked out has degree (sum of in-degree and out-degree) at least 1. This random masking is only done once, that is, it is fixed when evaluating the models. For each validation grounded scene graph, we conduct the completion 200 times, and report the Hit Rate @ K (**HR@K**) and mean accuracy (**mA**) values. For each masked grounded scene graph, to calculate the HR@K, we first build a node label histogram from the 200 completions over the 150 object categories, and then keep the predictions from the K most frequently predicted node categories. We assign a score of 1 if there is one prediction from the kept prediction set matches the ground-truth node label, and a score of 0 otherwise. If there are multiple categories on the boundary when selecting the top K predicted categories, we randomly select some of them to make it exact K categories. Accuracy is defined as the ratio of the correct predictions (matched to the ground-truth) among the 200 predictions. We then respectively take the average value of HR@K and accuracy over the

Table 6: **Grounded scene graph classification evaluation results** in the PredCls and SGCls settings.

| Method | PredCls Mean Triplet Acc ↑ | SGCls Object Acc ↑ | SGCls Mean Triplet Acc ↑ | FID ↓ |
|---|---|---|---|---|
| Ground-Truth Images | 30.65 | 70.31 | 15.77 | 0.0 |
| LayoutDiffusion | **27.85** | **56.79** | **10.04** | 15.73 |
| Object-ControlNet | 26.68 | 55.05 | 8.35 | **15.32** |
| Relation-ControlNet | 27.08 | 47.94 | 9.07 | 15.99 |

whole validation set to get the final validation HR@K and mA scores. Tab. 5 shows the HR@1/10/50/100 and mA results on the single node label completion task. As the results suggested, on all evaluation metrics, our DiffuseSG is consistently better than the DiGress (Vignac et al., 2022) baseline.

**Single Edge Label Completion.** Similar to the above single node label completion task, we conduct another single edge label completion task, where one edge label is masked out per validation grounded scene graph. The experiment setting and evaluation metrics are the same as the single node label completion setting. For each partially masked grounded scene graph, we predict the edge label 200 times. Tab. 5 shows the HR@1/10/25/50 and mA results. Again, our DiffuseSG is consistently better than DiGress on all evaluation metrics.

**Single Bounding Box Completion.** Some qualitative examples from DiffuseSG on the single bounding box completion task are shown in Fig. 5 (more in Appendix D.2), where one bounding box location is masked out given a validation grounded scene graph, with all other information untouched. The node whose bounding box is masked out has degree at least 1. Note that neither the image nor any image feature is given to the model for the completion task; the image is only for visualization. As seen, DiffuseSG can complete the bounding box in reasonable locations.

### 5.3 Conditional Image Generator Discussion

As described in Sec. 4.2, we have two conditional image generators: a layout-to-image model LayoutDiffusion (Zheng et al., 2023b) and a relation-aware model Relation-ControlNet. In this section, we are going to compare the image generation quality between these two generators. The COCO-Stuff dataset only has spatial relation types, which are less semantically meaningful because these relations can be simply decided from relative object bounding box locations. Thus we only use the VG dataset for the comparison.

#### 5.3.1 Grounded Scene Graph Classification Evaluation

The grounded scene graph classification evaluation is to measure whether the generated images follow the condition control, that is, the object labels and locations, and the relation labels. For each generated image, we use trained grounded scene graph classification models to classify grounded scene graphs under two settings: (1) PredCls, where given the ground-truth object labels and bounding box locations, to classify the relation labels; and (2) SGCls, where given the object bounding box locations, to classify both object labels and relation labels. The PredCls setting is to evaluate whether the generated images contain the input relations at appropriate locations, while the SGCls setting is to check whether the objects and relations are generated properly as a whole at the given locations.

We use classification accuracies as the evaluation metrics. We calculate two types of accuracies: one for object labels, and one for triplet labels (the subject, relation, object labels in the <`subject`, `relation`, `object`> triplets). Given the long-tailed nature of the relation labels in the VG dataset, while calculating the triplet label accuracy, we first take the average of the accuracies for each relation category, and then average across all relation categories, which weighs each relation category equally. The triplet accuracy is calculated under both the PredCls and SGCls settings while the object accuracy is calculated only under SGCls, since all the object information is given in PredCls.

Table 7: **Relation control evaluation results** under the PredCls and SGCls settings.

| | PredCls | SGCls | |
| Method | Mean Triplet Acc ↑ | Object Acc ↑ | Mean Triplet Acc ↑ |
|---|---|---|---|
| LayoutDiffusion | 21.10 | **58.32** | 9.10 |
| Object-ControlNet | 17.88 | 56.28 | 7.95 |
| Relation-ControlNet | **24.23** | 55.21 | **10.95** |

The grounded scene graph classification models used are the MOTIFS-SUM-TDE models [6] (Tang et al., 2020), given their popularity, easy access, and reasonable performance. Besides LayoutDiffusion and Relation-ControlNet, we also consider another variant of ControlNet as a baseline, named Object-ControlNet. The differences between Object-ControlNet and Relation-ControlNet are that in Object-ControlNet, the control input contains only the object information and the text prompt includes all English words of object labels (allowing repetitions). This can be considered as an ablation of Relation-ControlNet, which omits relation information. Note, both LayoutDiffusion and Object-ControlNet generate images conditioned on layout, while Relation-ControlNet generates images from grounded scene graphs.

Given an image generator, we use the $5,000$ grounded scene graphs from the VG's validation set, each to generate 5 images, resulting in $25,000$ images. The accuracy results are presented in Tab. 6[7], where the object label accuracy is denoted as **Object Acc** and the triplet label accuracy is denoted as **Mean Triplet Acc**. We also report the accuracies of the grounded scene graph classifiers on VG's ground-truth validation images, to show the upper bounds of the accuracy values. We also present the FID scores of the generated images to show the general image generation quality.

As the results suggested, the images generated from all conditional image generators have comparable general image quality (similar FIDs). However, different image generators show different control abilities. Though both LayoutDiffusion and Object-ControlNet receive the same object control information: object labels and bounding box locations, the Object-ControlNet shows worse object renderings, which also results in worse relation renderings. We believe this result difference is due to the different model architectures and training strategies between these two models. The Relation-ControlNet includes relation labels as an additional control input compared to Object-ControlNet, thus it has better triplet accuracies even with worse object accuracy. The worse object renderings of Relation-ControlNet is reasonable. It is a common observation that it is more difficult to generate images properly with more condition inputs. Note that even the triplet accuracies of Relation-ControlNet are better than those of Object-ControlNet, the differences are marginal (0.4 under PredCls and 0.72 under SGCls), and they are still worse than those of LayoutDiffusion. This is due to the fact that in VG, many relations can be inferred from bounding box positions. This is an artifact of the dataset and not a general problem at hand.

### 5.3.2 Relation Control Evaluation

To show the benefits of Relation-ControlNet, we build an evaluation set specifically designed for relation control evaluation. Among the 50 VG relation categories, we choose the subset containing `carrying`, `eating`, `holding`, `laying on`, `looking at`, `lying on`, `playing`, `riding`, `sitting on`, `standing on`, `using`, and `watching`. These are the relations which can not be easily determined by relative bounding box locations. In the validation set, we find the <`subject`, `relation`, `object`> triplets containing those relation categories. For each such triplet, we find its closest triplet inside the validation set. The closest triplet is to have the same subject and object labels, but a different relation label as the original one. The relative bounding box position of the subject and the object of the closest triplet is also the same as the original one[8]. We discard the triplets for which we can not find the closest counterpart. If there are more than one such closest triplets, we choose the one which has the most similar subject and object bounding box aspect ratios as

---

[6] The model checkpoints are downloaded from `https://github.com/KaihuaTang/Scene-Graph-Benchmark.pytorch`.

[7] As discussed in Appendix C.1, the result calculation in this Table is actually inferior to LayoutDiffusion. However, it achieves the best accuracies compared to both Object-ControlNet and Relation-ControlNet.

[8] The relative bounding box positions are defined as the same way used in the COCO-Stuff dataset.

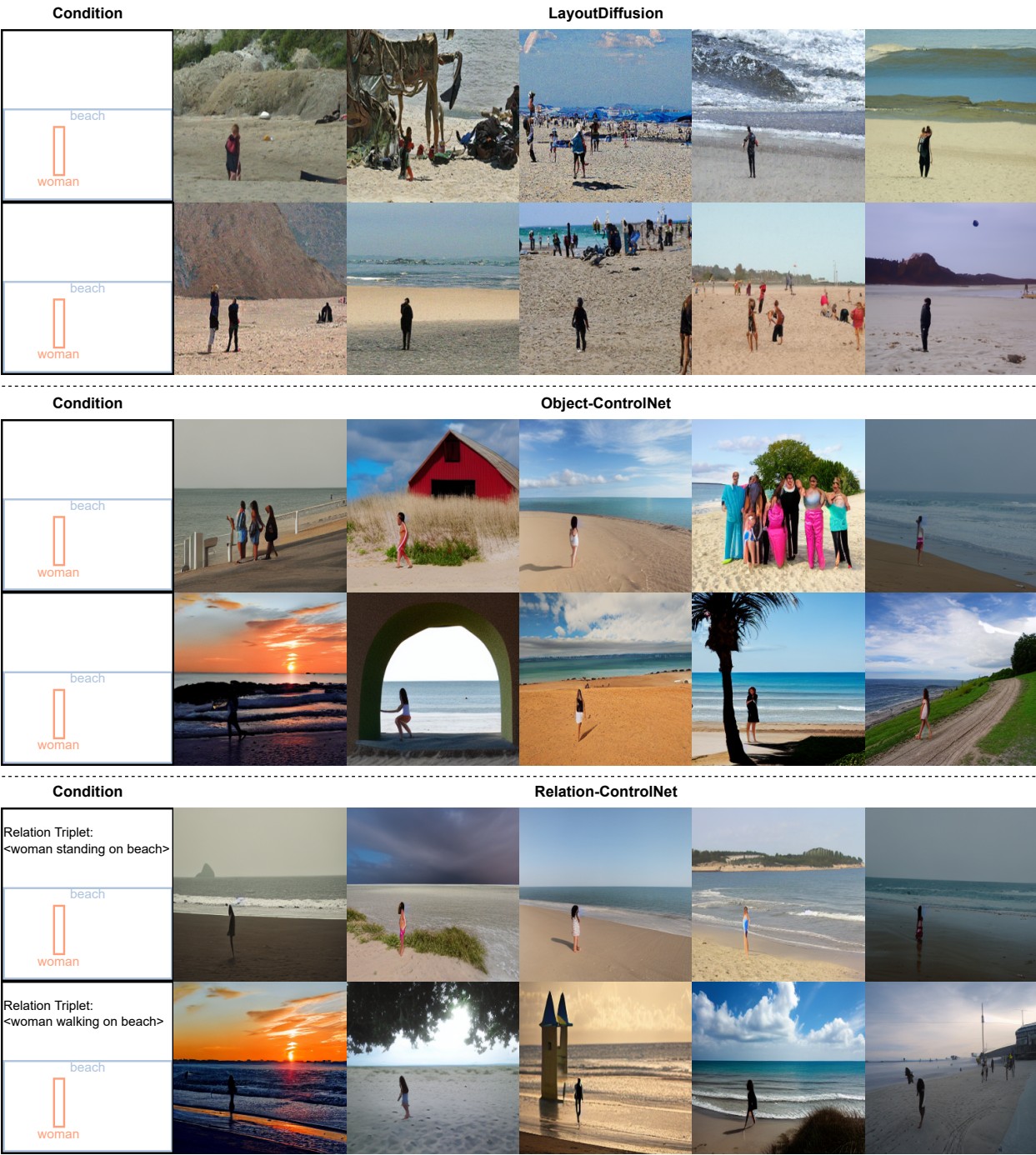

Figure 6: **Relation control evaluation qualitative results.** The same layout condition containing an object `woman` and an object `beach` with their corresponding bounding box locations is fed to all the models. Additionally, the relation information is given to the Relation-ControlNet model. We draw 10 image samples from both LayoutDiffusion and Object-ControlNet, while with Relation-ControlNet, we draw 5 samples on the relations <`woman standing on beach`> and <`woman walking on beach`> respectively.

the original triplet. We then swap the relation labels between the triplet and its closest counterpart. We keep all the triplets, their closest triplets, and the triplets with swapped relations. This process gives us 328

Table 8: **Grounded scene graph detection (SGDet) results** of SGTR.

| Data Setting | mR@50 | mR@100 | R@50 | R@100 | Head | Body | Tail |
|---|---|---|---|---|---|---|---|
| Original | 11.9 | 16.0 | **23.7** | **27.0** | **26.6** | 19.8 | 9.0 |
| Additional | **12.7** | **16.7** | 23.6 | 26.8 | 26.4 | **20.7** | **9.7** |

triplets. The swapping procedure guarantees that for the same pair of subject and object, there exist two different relation types, which is agnostic to the object-only conditioned models, but important to the relation conditioned model. For each triplet, we generate 5 images from LayoutDiffusion, Object-ControlNet, and Relation-ControlNet respectively. The corresponding object and triplet accuracies are reported in Tab. 7.

This setting shows the benefits of Relation-ControlNet. As indicated in the results, though Relation-ControlNet is worse than LayoutDiffusion and Object-ControlNet in terms of object accuracy, its triplet accuracies are significantly better than the other two. This is because LayoutDiffusion and Object-ControlNet can not take the relation label as input but Relation-ControlNet can. As the qualitative examples shown in Fig. 6, when the relation information is inputted to the model, Relation-ControlNet can constantly render the appropriate postures for the `woman`, either standing or walking depending on the condition. However, for both LayoutDiffusion and Object-ControlNet, since these models can not read in the relation information, the models have to render the postures of the `woman` via their own decisions, which may be standing, walking, or even bending the knees. Though our Relation-ControlNet is not perfect, it implies the usefulness of generating grounded scene graphs. Generating grounded scene graphs, which contain object labels and locations along with relation labels, can provide more controllability for image generation.

Relation-ControlNet is designed for a harder task, where the relation information is also part of the condition. However, given the inferior performance of Relation-ControlNet in the real validation setting (Tab. 6) and limitations of current datasets (many of the relations in VG can be easily determined by relative bounding box positions), we use LayoutDiffusion as the conditional image generator for experiments that follow.

## 5.4 Grounded Scene Graph Detection Evaluation

We take the DiffuseSG trained on VG, and pair it with the trained LayoutDiffusion model (Zheng et al., 2023b) to form $5,000$ grounded scene graph - image pairs, treated as additional training data for the downstream grounded scene graph detection (SGDet) task: given an image, detecting the object labels and locations, as well as the relation labels. Those additional grounded scene graphs only contain body relations as defined in Li et al. (2021a). Detailed process is in Appendix C.2.

We use the SGTR model (Li et al., 2022a), as an example, to show the value of our generated grounded scene graph - image pairs. We train SGTR for the SGDet task under two training data settings: (1) Original, where the training data is the original Visual Genome training data in Li et al. (2022a); (2) Additional, where besides the original training data, we add in our generated $5,000$ grounded scene graph - image pairs. Note that these two settings only differ in the training data; both validation and testing data is still the original data for the SGDet task. For both data settings, we train SGTR 4 times and the averaged test results are reported in Tab. 8, where the model for testing is selected via best mR@100 on the validation set. We report results on mean Recalls (mR@50 and mR@100), Recalls (R@50 and R@100), and mR@100 on the head, body, and tail relation partitions (respectively indicated as Head, Body, and Tail in the Table).

Comparing our Additional results with the Original ones, we can see that our generated grounded scene graph - image pairs do have value, which brings improved results on Body and Tail, and comparable results on Head, which results in increased results on mean Recalls and comparable results on Recalls. As suggested in Tang et al. (2020), mean Recall is a better evaluation metric than Recall for the grounded scene graph detection task, because it is less biased to the dominant relation classes. Note that although the generated grounded scene graphs in the additional training data only contain body relations, the Tail results also get improved. This is reasonable, because grounded scene graph is a structure, increasing the confidence of some part of the structural prediction will increase the confidence of other part as well, especially for the tail relations, where the prediction confidence is usually low.

## 6 Conclusion

In this work, we propose a novel framework for joint grounded scene graph - image generation. As part of this, we propose DiffuseSG, a diffusion-based model for generating grounded scene graphs that adeptly handles mixed discrete and continuous attributes. DiffuseSG demonstrates superior performance on both unconditional grounded scene graph generation and conditional grounded scene graph completion tasks. By pairing DiffuseSG with a conditional image generation model, the joint grounded scene graph - image pair distribution can be obtained. We illustrate the benefits of DiffuseSG both on its own and as part of joint grounded scene graph - image generation. In the future, we are interested in modeling the joint distribution with a single model.

**Acknowledgments**

This work was funded, in part, by the Vector Institute for AI, Canada CIFAR AI Chairs, NSERC CRC, and NSERC DGs. Resources used in preparing this research were provided, in part, by the Province of Ontario, the Government of Canada through CIFAR, the Digital Research Alliance of Canada `alliance.can.ca`, companies sponsoring the Vector Institute, and Advanced Research Computing at the University of British Columbia. Additional hardware support was provided by John R. Evans Leaders Fund CFI grant and Compute Canada under the Resource Allocation Competition award. Qi Yan is supported by the UBC Four Year Doctoral Fellowship.

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

# A    Implementation Details

## A.1    Discrete Input Encodings

To effectively handle the discrete attributes, we implement the following encoding methods.

**Scalar.**  Similar to prior works (Ho et al., 2020; Song et al., 2021; Karras et al., 2022; Jo et al., 2022), we use zero-based indexing and map the scalar value to the range $[-1, 1]$. Specifically, during training, denoting an integer label of an attribute as $n$ and the number of categories of the attribute as $m$, where $n \in \{0, 1, \dots, m-1\}$, the scalar representation becomes $\frac{2n}{m-1} - 1$. During sampling, we first split the interval $[-1, 1]$ into equal-sized bins in accordance with the number of node or edge categories. We then decode the continuous-valued network output into a discrete label based on the bin into which the output value falls.

**Binary-Bit.**  Following Chen et al. (2022), we first convert the zero-based indexed integer node/edge attribute into binary bits, and then remap the 0/1 bit values to -1/1 for improved training stability. During sampling, we binarize each value in the network output based on its sign. That is, a positive value is interpreted as 1, and a negative value as 0. We then convert the binary representation back into an integer node or edge label.

**One-Hot.**  We remap the 0/1 values in the one-hot encoding of the original integer node/edge label to -1/1. During sampling, we take the argmax value of the network output to obtain the categorical label.

## A.2    DiffuseSG Network Architecture

Our proposed *graph transformer* has two essential components: (1) shifted-window attention mechanism, and (2) downsampling/upsampling layers. In the case of graphs comprising $n$ nodes, their adjacency matrices, incorporating both node and edge attributes, are conceptualized as high-order tensors with $n \times n$ entries. To handle graphs of varying sizes, we standardize the size of these adjacency matrices through padding. Consequently, for different datasets, we accordingly adjust the design parameters to ensure the network is proportionate and suitable for the specific requirements of each dataset.

**Shifted-Window Attention.**  We adopt the shifted window attention technique from Liu et al. (2021), which partitions the original grid-like feature map into smaller subregions. Within these subregions, local message passing is executed using self-attention mechanisms. Additionally, the windows are interleavedly shifted, facilitating cross-window message passing, thereby enhancing the overall efficiency and effectiveness of the feature extraction process.

**Downsampling/Upsampling Layer.**  We incorporate channel mixing-based downsampling/upsampling operators to effectively diminish or augment the size of the feature map, thereby constructing hierarchical representations. During downsampling, the feature map is divided into four segments based on the parity of the row and column indices, followed by a concatenation process along the channel, which serves to reduce the dimensions of height and width. The upsampling process performs the inverse operations. It initially splits the tensors along the channel and then reshapes them, effectively reversing the process conducted in downsampling. The downsampling and upsampling layers are visualized in Fig. 7. We also implement one MLP layer right after each downsampling/upsampling layer. In line with the widely recognized U-Net architecture (Song et al., 2021; Karras et al., 2022), our approach also integrates skip-connections for tensors of identical sizes to enhance the network capacity.

The crucial design parameters of our model are detailed in Tab. 9. It is important to note that within the Down/Up block layers, the initial blocks do not utilize downsampling/upsampling operations. For instance, in the context of the Visual Genome dataset, we effectively implement 3 downsampling layers, which leads to the successive alteration of the feature map dimensions as $64 \to 32 \to 16 \to 8$. In this setup, we opt for a window size of 8, ensuring that the receptive field is sufficiently large to facilitate effective message passing between each pair of nodes. While on COCO-Stuff, we employ 2 downsampling layers, resulting the feature map dimensions as $40 \to 20 \to 10$, and thus the window size is set to 10.

**MLP Prediction Head.**  The node/edge attribute MLP prediction head is implemented as two linear layers with a GELU (Hendrycks & Gimpel, 2016) operation injected in between.

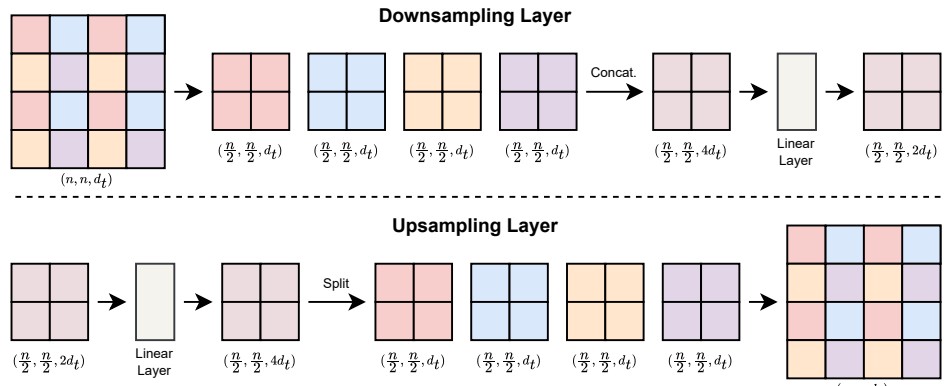

Figure 7: **Visualization** of the downsampling and upsampling layers.

Table 9: **Architecture details** of our graph transformer.

| Hyperparameter | VG | COCO-Stuff |
|---|---|---|
| Full tensor size $(n \times n)$ | $64 \times 64$ | $40 \times 40$ |
| Down block attention layers | $[1, 1, 3, 1]$ | $[1, 2, 6]$ |
| Up block attention layers | $[1, 1, 3, 1]$ | $[1, 2, 6]$ |
| Number of attention heads | $[3, 6, 12, 24]$ | $[3, 6, 12]$ |
| Window size | 8 | 10 |
| Token dimension | 96 | 96 |
| Feedforward layer dimension | 384 | 384 |

### A.3 DiffuseSG Diffusion Modeling Details

To ensure the stable training of our diffusion model, we adopt a framework based on the stochastic differential equation (SDE), as proposed in Song et al. (2021). Additionally, we incorporate a variety of training techniques that have been proven effective in image generation contexts: network preconditioning (Karras et al., 2022), self-conditioning (Chen et al., 2022) and exponential moving average (EMA). For network training, we employ the hyperparameters specified for the ImageNet-64 dataset in Karras et al. (2022) for preconditioning purposes; a detailed explanation of these parameters can be found therein. We use Adam optimizer and learning rate being 0.0002. The EMA coefficients used for evaluation are 0.9999 and 0.999 on the Visual Genome and COCO-Stuff datasets respectively.

The pseudocode of our sampling algorithm is presented in Algorithm 2, which follows the stochastic sampler in Karras et al. (2022) but with the additional self-conditioning (Chen et al., 2022) technique. In the algorithm, $D_\theta$ is the denoising network, $\widetilde{\boldsymbol{S}}^{(t)}$ is the generated grounded scene graph at step $t$, and $\boldsymbol{I}$ is the identity matrix. The associated parameters are detailed in Tab. 10. We opt for $T = 256$ sampling steps to expedite the sampling process, as opposed to the original $1,000$ steps used in the DDPM framework (Ho et al., 2020). In Algorithm 2, with a slight abuse of notations for simplicity, we consider the generated grounded scene graph $(\widetilde{\boldsymbol{S}}, \widehat{\boldsymbol{S}}_{\text{sc}})$, which comprises tuples of node and edge attributes, as a singular tensor, allowing for straightforward addition or subtraction operations. Practically, this is implemented through separate operations on the node and edge tensors.

### A.4 Relation-ControlNet and Object-ControlNet Training Details

Following Zhang et al. (2023b), we use Stable Diffusion as an instantiation of the ControlNet architecture. We use Stable Diffusion V1.5. To train both Relation-ControlNet and Object-ControlNet, we use Adam optimizer with $\beta_1$ being 0.9, $\beta_2$ being 0.999, and weight decay being 0.01; a constant learning rate 0.00001 is

---

**Algorithm 2** DiffuseSG Sampler.

---

**Require:** $D_\theta, T, \{t_i\}_{i=0}^T, \{\gamma_i\}_{i=0}^{T-1}$.

1: **sample** $\widetilde{\boldsymbol{S}}^{(0)} \sim \mathcal{N}(\boldsymbol{0}, t_0^2 \boldsymbol{I}), \widehat{\boldsymbol{S}}_{\text{sc}}^{(0)} = \boldsymbol{0}$.

2: **for** $i = 0$ to $T - 1$ **do**

3:     **sample** $\boldsymbol{\epsilon} \sim \mathcal{N}(\boldsymbol{0}, S_{\text{noise}}^2 \boldsymbol{I})$

4:     $\hat{t}_i \leftarrow (1 + \gamma_i) t_i$

5:     $\widetilde{\boldsymbol{S}}^{(\hat{i})} \leftarrow \widetilde{\boldsymbol{S}}^{(i)} + \sqrt{\hat{t}_i^2 - t_i^2} \boldsymbol{\epsilon}$

6:     $\widehat{\boldsymbol{S}}_{\text{sc}}^{(\hat{i})} \leftarrow D_\theta(\widetilde{\boldsymbol{S}}^{(\hat{i})}, \widehat{\boldsymbol{S}}_{\text{sc}}^{(i)}, \hat{t}_i)$

7:     $\boldsymbol{d}_i \leftarrow (\widetilde{\boldsymbol{S}}^{(\hat{i})} - \widehat{\boldsymbol{S}}_{\text{sc}}^{(\hat{i})})/\hat{t}_i$

8:     $\widetilde{\boldsymbol{S}}^{(i+1)} \leftarrow \widetilde{\boldsymbol{S}}^{(\hat{i})} + (t_{i+1} - \hat{t}_i) \boldsymbol{d}_i$

9:     $\widehat{\boldsymbol{S}}_{\text{sc}}^{(i+1)} \leftarrow D_\theta(\widetilde{\boldsymbol{S}}^{(i+1)}, \widehat{\boldsymbol{S}}_{\text{sc}}^{(\hat{i})}, t_{i+1})$

10:     $\boldsymbol{d}_i' \leftarrow (\widetilde{\boldsymbol{S}}^{(i+1)} - \widehat{\boldsymbol{S}}_{\text{sc}}^{(i+1)})/t_{i+1}$

11:     $\widetilde{\boldsymbol{S}}^{(i+1)} \leftarrow \widetilde{\boldsymbol{S}}^{(i)} + \frac{1}{2}(t_{i+1} - \hat{t}_i)(\boldsymbol{d}_i + \boldsymbol{d}_i')$

12: **end for**

13: **return** $\widetilde{\boldsymbol{S}}^{(T)}$

---

Table 10: **Sampling parameters** in the denoising process.

$$\sigma_{\min} = 0.002, \sigma_{\max} = 80, \rho = 7$$
$$S_{\text{tmin}} = 0.05, S_{\text{tmax}} = 50, S_{\text{noise}} = 1.003, S_{\text{churn}} = 40, T = 256$$
$$t_i = (\sigma_{\max}^{\frac{1}{\rho}} + \frac{i}{T-1}(\sigma_{\min}^{\frac{1}{\rho}} - \sigma_{\max}^{\frac{1}{\rho}}))^\rho$$
$$\gamma_i = \mathbf{1}_{S_{\text{tmin}} \leq t_i \leq S_{\text{tmax}}} \cdot \min(\frac{S_{\text{churn}}}{T}, \sqrt{2} - 1)$$

used to train the models. Both models are trained for 200 epochs with a batch size of 120. Following Zhang et al. (2023b), during training, the text prompts are randomly replaced with empty strings at a chance of 50%. Images are generated in the resolution of $256 \times 256$.

## B  D3PM Baseline

Our D3PM (Austin et al., 2021) baseline is based on the image generation model on the CIFAR-10 dataset (Krizhevsky et al., 2009). Given a scene graph $\boldsymbol{S} = (\boldsymbol{V}, \boldsymbol{E})$[9] with $n$ nodes, where $\boldsymbol{V} \in \mathbb{N}^n$ is the node vector containing the integer node labels, and $\boldsymbol{E} \in \mathbb{N}^{n \times n}$ is the adjacency matrix containing the integer edge labels, the input to our D3PM baseline is represented as $\boldsymbol{Q} \in \mathbb{N}^{n \times n \times 3}$, where $\boldsymbol{Q}_{i,j} = [\boldsymbol{V}_i, \boldsymbol{V}_j, \boldsymbol{E}_{i,j}]$, $\forall i, j \in \{1, 2, \ldots, n\}$.

We use two separate discretized Gaussian transition matrices, one for the node category and one for the edge category, to add noise on $\boldsymbol{Q}$, resulting in the noised $\tilde{\boldsymbol{Q}}$. We then use a U-Net with two separate prediction heads, each implemented as two convolution layers with a sigmoid operation before each of the convolution layers, to respectively produce the logits of the denoised $\hat{\boldsymbol{V}}$ and $\hat{\boldsymbol{E}}$, which then form the logits of the denoised $\hat{\boldsymbol{Q}}$. We use the $L_{\lambda=0.001}$ (Eq. (5) in Austin et al. (2021)), calculated on the logits of $\hat{\boldsymbol{Q}}$, to train our D3PM baseline. We use Adam optimizer and learning rate being 0.00005 for training. We use $T = 1,000$ noising and denoising steps and take the model with EMA coefficient 0.9999 for evaluation.

The $\beta_t$ (in Eq. (8) in the Appendix of Austin et al. (2021)) of the discretized Gaussian transition matrix is increased linearly, for $t \in \{1, 2, \ldots, T\}$. On the Visual Genome dataset, we set $n$ to be 64, and $\beta_t$ is increased linearly from 0.0001 to 0.02 for both node and edge categories. On the COCO-Stuff dataset, $n$ is set to be 36. The $\beta_t$ is increased linearly from 0.0001 to 0.02 for the node category and from 0.04 to 0.1 for the edge category.

---

[9]We slightly abuse the notations here, specifically for the D3PM model, compared to the ones in the main text.

## C  Evaluation Details

### C.1  Grounded Scene Graph Classification Evaluation on LayoutDiffusion

The LayoutDiffusion model checkpoint that we used is trained on a version of the VG dataset annotation which has 178 object categories. This is slightly different from the VG dataset annotation that the grounded scene graph classification models are trained on, which contains 150 object categories. However, between these two versions of VG annotations, there are 131 object categories in common. Thus when generating images from LayoutDiffusion for the grounded scene graph classification evaluation, we only keep the objects whose labels are in the common category set. Specifically, given a ground-truth VG validation grounded scene graph, we keep the objects in the common category set and discard others, and then generate the corresponding images. But when calculating the grounded scene graph classification accuracy scores, we still use the ground-truth grounded scene graph without any object filtering. Though this setting is inferior to LayoutDiffusion, it still achieves better accuracies than Object-ControlNet and Relation-ControlNet, as shown in Tab. 6.

When building the evaluation set for the relation control evaluation, we make sure that all the subject and object labels are in the common category set.

### C.2  Generating Additional Training Data for the Grounded Scene Graph Detection Task

We take our DiffuseSG model trained on the Visual Genome dataset, let it generate a set of grounded scene graphs which only contain relations falling into the body partition (as defined in Li et al. (2021a)), and then use the pretrained (on VG, with resolution $256 \times 256$) LayoutDiffusion model (Zheng et al., 2023b) to form the grounded scene graph - image pairs. Since there exists some node label set discrepancy between the respective VG annotations used to train the LayoutDiffusion model (178 node categories) and our DiffuseSG model (150 node categories). When forming the grounded scene graph - image pairs, we discard the nodes whose labels are not in the common category set (131 node categories) and their related edges. We randomly choose 5,000 such generated pairs, where node numbers are restricted to be less than 10, as additional training data to train the SGTR model (Li et al., 2022a) on the grounded scene graph detection task: given an image, detecting a grounded scene graph (node labels and bounding box locations, and edge labels) from it. We guarantee that for those randomly chosen grounded scene graphs, each of them has at least one edge.

The motivations of why we generating the grounded scene graphs only containing body relations are as follows. First, for the grounded scene graph detection task, there are already many training instances for the head classes, so generating additional head relations may not be beneficial at all. Second, for our grounded scene graph generation task, since the training data for the tail relations is limited, our grounded scene graph generation model may not be able to model the tail class distribution well.

## D  More Qualitative Results

### D.1  Grounded Scene Graph - Image Pair Generation

More qualitative results of grounded scene graph - image pair generation are shown in Figs. 8 and 9 (Visual Genome) and Figs. 10 and 11 (COCO-Stuff). Grounded scene graphs including the bounding box locations are generated by DiffuseSG and the corresponding images (in resolution $256 \times 256$) are produced by the pretrained LayoutDiffusion model (Zheng et al., 2023b). Note that on the Visual Genome dataset, since there exists some node label set discrepancy between the annotations used to train our DiffuseSG model and the LayoutDiffusion model, we only visualize the grounded scene graphs whose node labels are all in the common node label set (131 node categories).

### D.2  Single Bounding Box Completion

More qualitative results of our DiffuseSG on the Visual Genome validation set are shown in Fig. 12. The left figure shows the input grounded scene graph, where only the edges and corresponding node labels are

shown. The blue node's bounding box has been masked out. The middle figure shows the untouched (input) bounding boxes with labels in red, the one masked out in blue, along with the corresponding ground-truth image. The right figure shows our generated bounding box heatmap in white along with the target ground-truth bounding box (to be completed) in blue. The heatmap is obtained via generating the bounding box 100 times; the whiter the area, the more overlap at the location. Note that neither the image nor any image feature is given to the model for the completion task; the image is only for visualization.

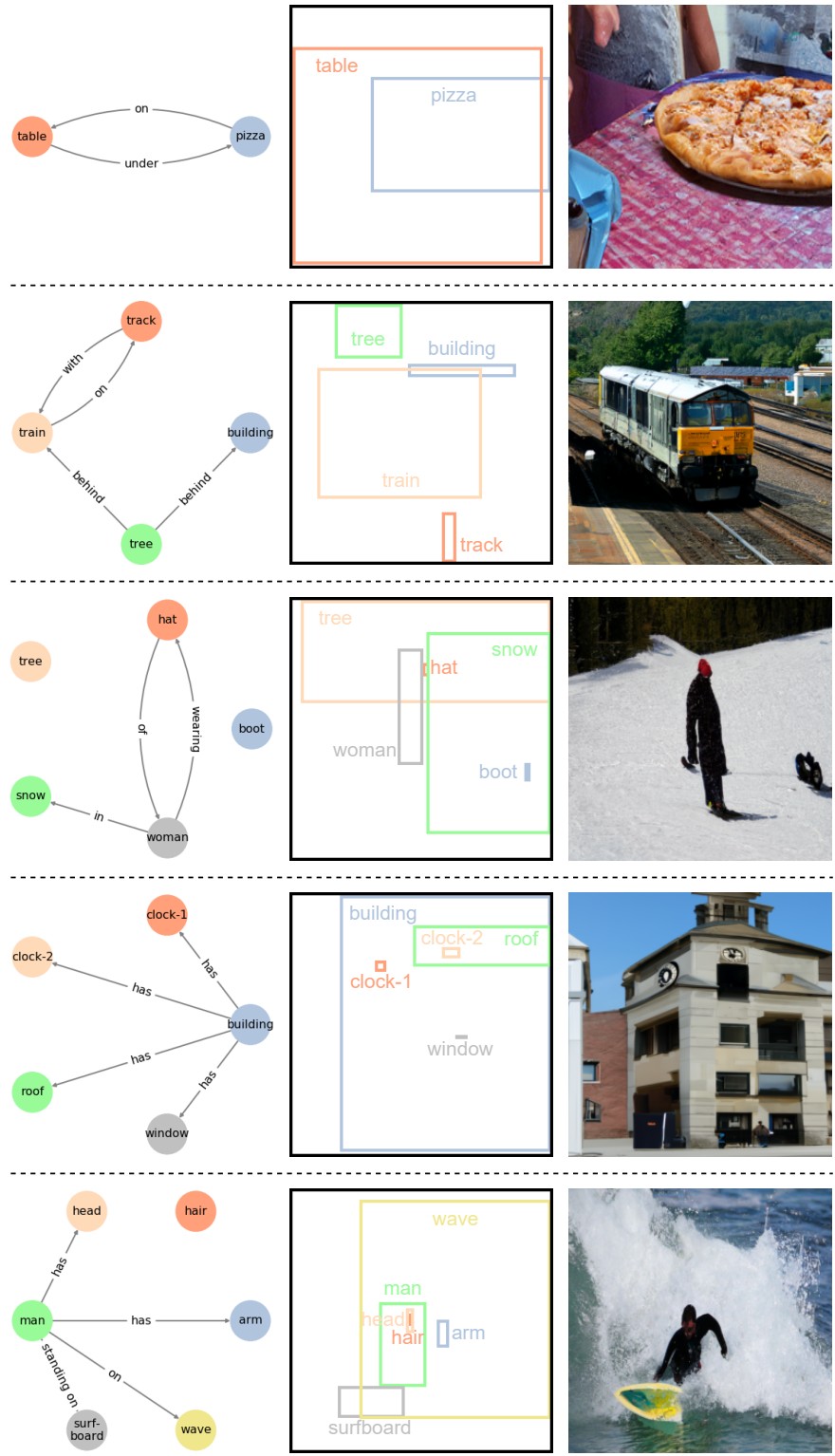

Figure 8: **Grounded scene graph - image pair generation** qualitative results on the Visual Genome dataset.

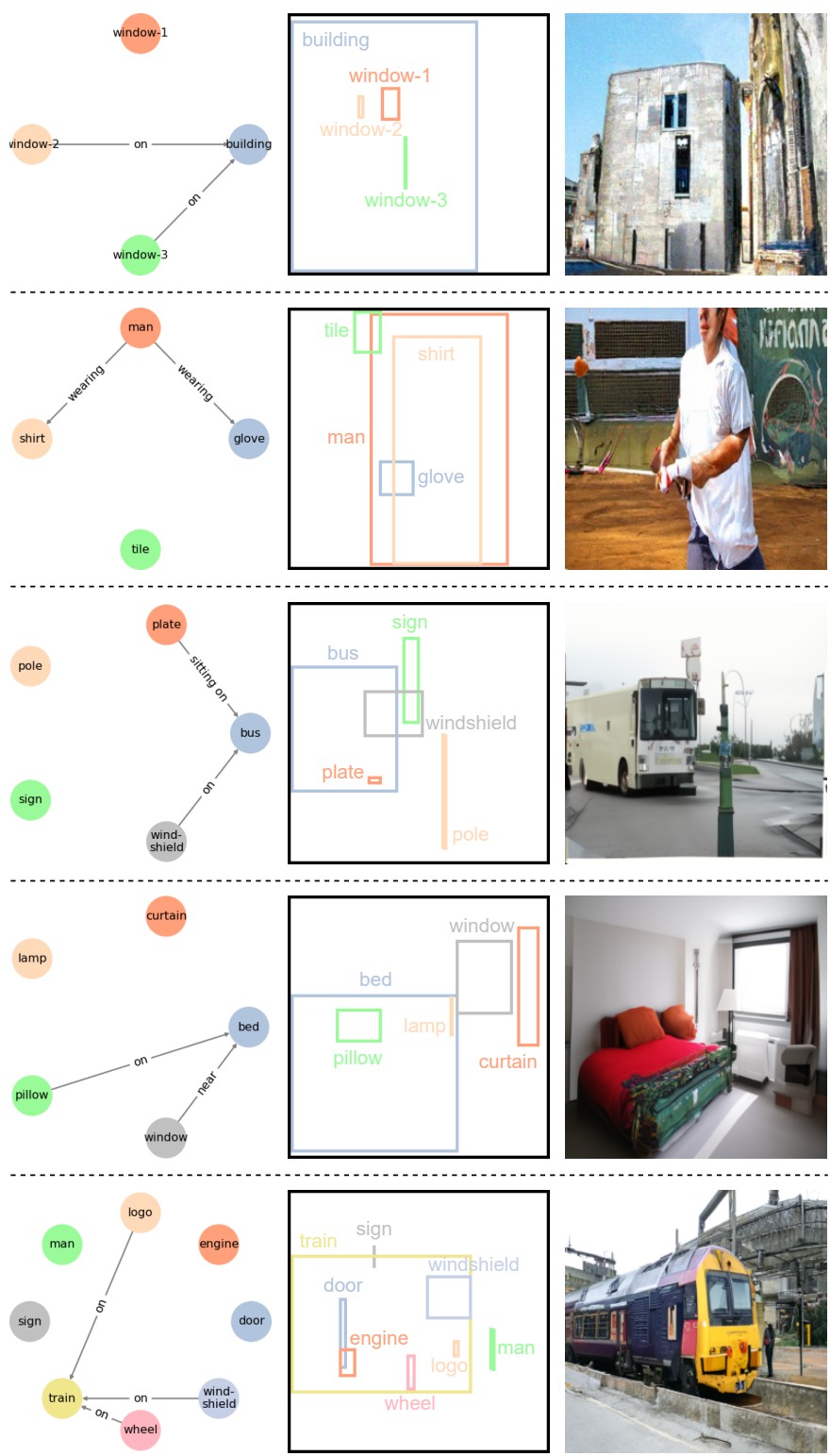

Figure 9: **Grounded scene graph - image pair generation** qualitative results on the Visual Genome dataset.

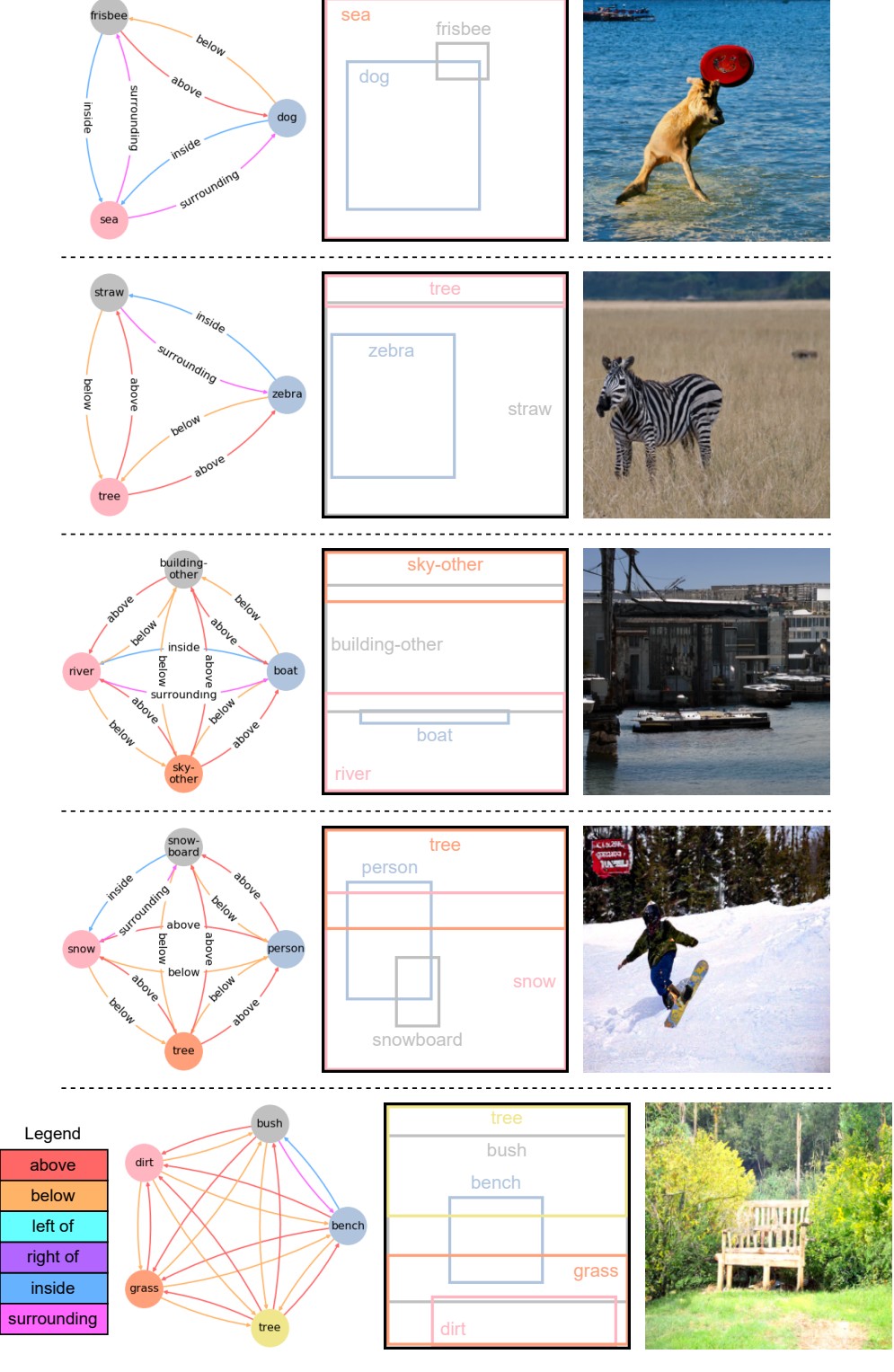

Figure 10: **Grounded scene graph - image pair generation** qualitative results on the COCO-Stuff dataset.

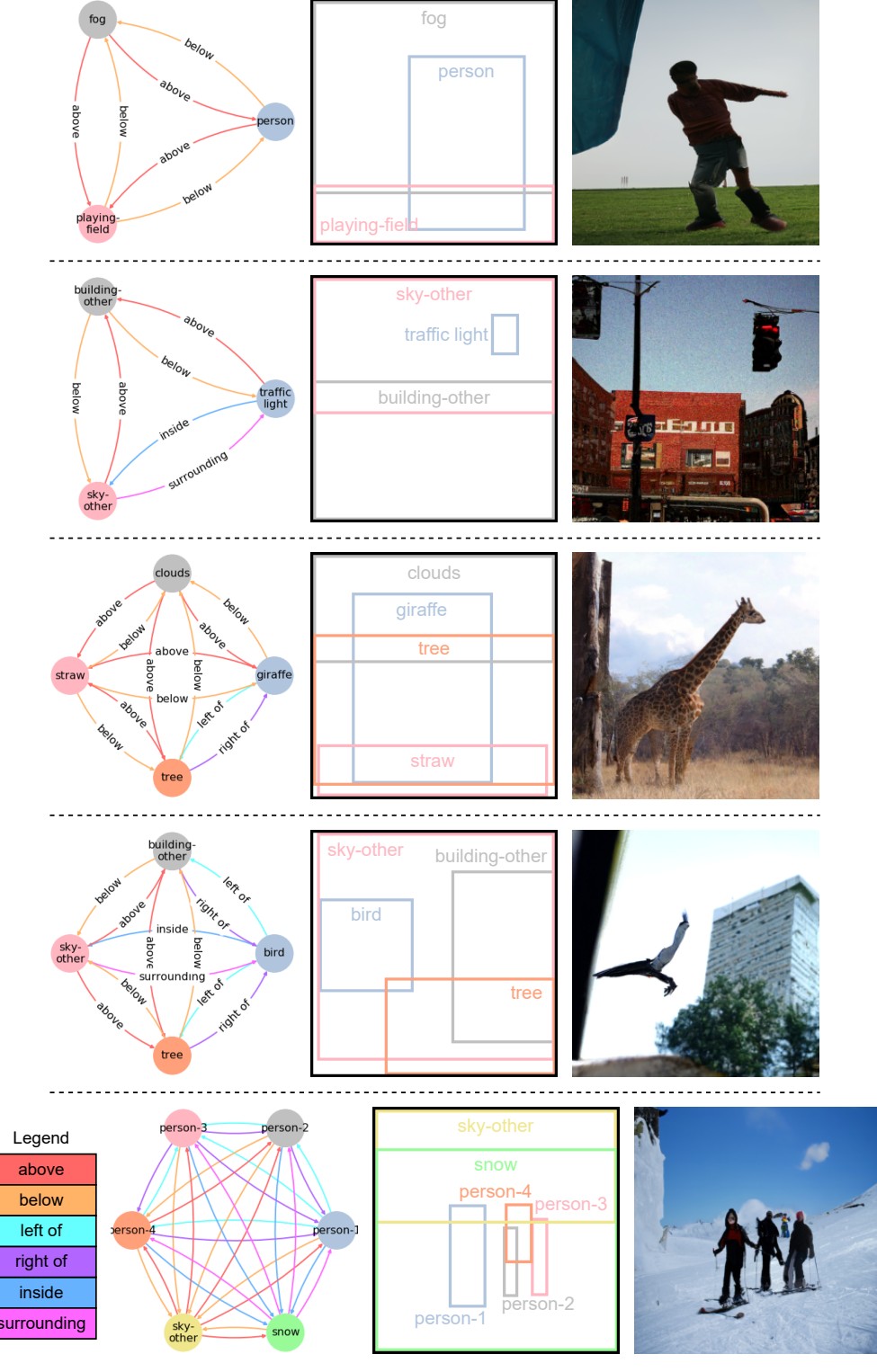

Figure 11: **Grounded scene graph - image pair generation** qualitative results on the COCO-Stuff dataset.

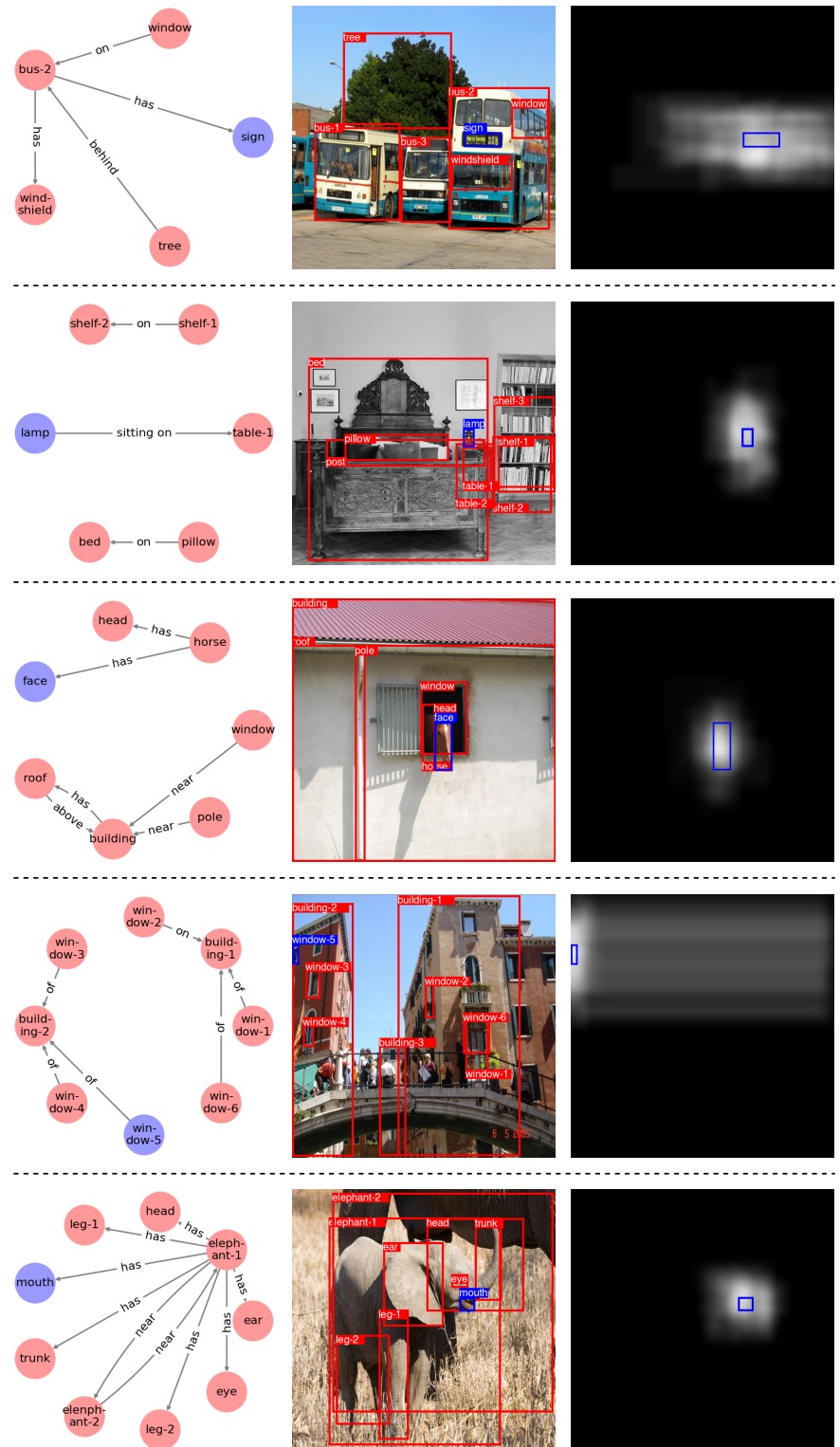

Figure 12: **Single bounding box completion** qualitative results on the Visual Genome validation set.

