# OpenReview forum: "Joint Generative Modeling of Grounded Scene Graphs and Images via Diffusion Models"
_TMLR — Accepted by TMLR_

### Review · Reviewer_3NQy · 2025-04-06

**Summary Of Contributions:**

This paper proposed DiffuseSG, a diffusion-based unconditional graph generative model that generates high-quality scene graphs. Specifically, the proposed method is designed to jointly denoise the object nodes and object relation edges via a graph-transformer-based denoising network with two predictive heads. The contributions of this paper is three-folds:

1. To the best of my knowledge, this paper is the first attempt that leverage diffusion-based denoising model in the scene graph generation task.
2. This paper explores the network design, categorical variable encoding schemes, and  bounding box IoU penalty design in training diffusion-based scene graph generative models.
3. The proposed DiffuseSG enables scene-graph-to-image generation by providing spatial conditioning controls to existing pre-trained vision generative models.

**Audience:**

Yes

**Claims And Evidence:**

Yes

**Requested Changes:**

See Concern 1-3.

**Strengths And Weaknesses:**

In general, this paper is well-written. The motivation is clear, and the logic flow is fluent. The notation system is clear self-contained. The scene graph generation task is an important task in vision generation, and the authors conduct a systematic research in the how well does diffusion-based method perform on this particular tasks.

The experiment sections are also admirable. The performance of DiffuseSG is comprehensively evaluated from both the graph generation quality aspect (Section 4.2) and the downstream scene-to-image generation aspects (Section 4.3). The Appendix sections provided sufficient details in reproducing the experiments.

-------
**Concern 1. Additional illustration in the training scheme.**

I recommend the authors to add a detailed algorithm panel for clarifying the training process of DiffuseSG, which is outlined in Setion 3.1.3.

**Concern 2. Additional illustration in the network designs.**

I recommend the authors to add a detailed diagram for illustrating the network architecture design of the proposed DiffuseSG, including the best practice of encoding scheme, graph embedding, and graph transformers (i.e., visualizing Appendix A).

**Concern 3. Additional illustration in conditional image generation.**

I recommend the authors to add a detailed diagram for the process of adapting DiffuseSG into pretrained Relation / Object-ControlNet (i.e., elaborating and visualizing Appendix A.4).

---

> ### Author Response · Authors · 2025-05-28
> **To Reviewer 3NQy**
>
> Dear Reviewer 3NQy,
>
> Thank you for your review and feedback. Below we address each of your concerns in turn.
>
> (1) Regarding illustration of the training scheme.
> - A detailed algorithm of the DiffuseSG’s training process is added into Section 3.1.3. The algorithm is titled “Algorithm 1 DiffuseSG Training Process”.
>
> (2) Regarding illustration of DiffuseSG’s design.
> - A detailed diagram illustrating the binary-bit encoding (best practice) of object and relation categories, graph embedding, and graph transformer is added into Section 3.1.3; the figure is labeled as Figure 2. The downsampling and upsampling layers are further visualized in Appendix A.2; the figure is labeled as Figure 7.
>
> (3) Regarding illustration of Relation-ControlNet or Object-ControlNet.
> - A diagram visualizing the Relation-ControlNet is added into Section 3.2.2; the figure is labeled as Figure 3. Given a grounded scene graph generated from DiffuseSG, we need to first compose the control input and text prompt, as described in Section 3.2.2, which are then fed into Relation-ControlNet or Object-ControlNet. The differences between Relation-ControlNet and Object-ControlNet are that: (1) in the control input, Object-ControlNet has only $C_o$ channels to represent the object locations; and (2) Object-ControlNet’s text prompt is a list of English words of all the object labels.

---

> > ### Comment · Reviewer_3NQy · 2025-05-29
> >
> > Dear Authors,
> >
> > The added algorithm illustrations are clear and informative. My previous concerns are well-addressed.
> >
> > Best,
> >
> > Reviewer 3NQy

---

### Review · Reviewer_nn1a · 2025-04-29

**Summary Of Contributions:**

This paper introduces a novel method called DiffuseSG, which effectively generates nodes, edges, and bounding boxes to construct scene graphs and layouts. It shows good empirical performance in scene graph and layout generation tasks compared to a range of existing methods. Additionally, its contributions are assessed across various related tasks, including conditional image generation, scene graph completion, and synthetic data generation to enhance scene graph detection.

**Audience:**

Yes

**Broader Impact Concerns:**

There is no concern on the ethical implications of this work from my perspective.

**Claims And Evidence:**

Yes

**Requested Changes:**

- The terms "scene graphs" and "layout" in this paper are somewhat confusing. It appears that the task of DiffuseSG blends elements of scene graph generation and layout generation. Its output includes node labels, bounding boxes, and edge labels (node relations), which differs from the typical definition of scene graphs in the literature, where only nodes and edges are included. This should be further clarified.
- Baselines such as SceneGraphGen, VarScene, D3PM, DiGress, do not have default bounding box output, while baselines such as BLT, LayoutDM, do not output relation labels. A table that lists these features would be helpful for readers to get a clear idea of the each baseline.
- Relation-ControlNet is an interesting method. A diagram to show the conditional image generation pipeline should be helpful to understand it.
- Table 1, experimental results of DiffuseSG with different input representations (scalar, one-hot) with the IoU loss are missing.

Questions
- LayoutDiffusion does not utilize relation labels by default. Are relation labels predicted by DiffuseSG used in your experiments of LayoutDiffusion-based conditional image generation?

**Strengths And Weaknesses:**

Strengths
- The proposed DiffuseSG method is novel, capable of generating nodes, edges, and bounding boxes, and demonstrates strong empirical performance in scene graph and layout generation tasks.
- A variety of evaluation setups were explored.
- Overall, the presentation of this paper is clear.

Weaknesses
- While this paper claims to introduce a novel joint scene graph-image generation framework, its primary contribution is in the scene graph and layout generation step. Its impact on scene graph-conditioned image generation seems limited, as it primarily adapts existing conditional image generation methods, such as LayoutDiffusion and ControlNet. These two processes remain quite separate rather than being truly "joint".
- From my perspective, the formulation of image generation conditioned on scene graphs is standard and does not constitute a significant contribution.
- Further clarification is needed in Section 3.1.3 regarding the technical novelty of DiffuseSG. It is unclear which aspects are novel and which are adapted from existing methods. An architecture diagram of DiffuseSG would be beneficial for understanding its technical innovations.

---

> ### Author Response · Authors · 2025-05-28
> **To Reviewer nn1a**
>
> Dear Reviewer nn1a,
>
> Thank you for your review and feedback. Below we address each of your raised points in turn.
>
> (1) Regarding the paper contributions.
> - We are doing joint grounded scene graph - image pair generation. To achieve this goal, we factorize the joint distribution into a grounded scene graph prior and a conditional distribution of images given the grounded scene graph. Though our conditional image generation methods seem standard, they constitute part of the joint grounded scene graph - image pair modeling.
>
> (2) Regarding the technical novelty of DiffuseSG.
> - We have added an architecture diagram of DiffuseSG (graph transformer) in Section 3.1.3, which is labeled as Figure 2. We also added a paragraph describing our technical contributions at the end of Section 3.1.3, which is pasted below.
> - “**Summary of Technical Contributions.** Though the shifted window attention is adopted from Liu et al. (2021), and the U-Net architecture is similar to other diffusion models like the ones in Song et al. (2021); Karras et al. (2022). Our novelties lie in: (1) the tensor representation of the grounded scene graph for DiffuseSG (a continuous diffusion model), (2) the separate read out layers (MLP prediction heads) for object and relation generation, (3) an additional bounding box IoU loss for training, and (4) exploration among different encoding mechanisms for object and relation categories.”
>
> (3) Regarding the confusion among three similar but distinct visual scene representations.
> - Now we are using three distinct terms, “layout”, “scene graph”, and “grounded scene graph” to describe the three visual scene representations.
> - **Layout**: containing only object labels and object bounding box locations.
> - **Scene graph**: containing only object labels and relation labels.
> - **Grounded scene graph**: containing all of object labels, object bounding box locations, and relation labels.
> - Our DiffuseSG generates grounded scene graphs. In the revised paper, we added a subsection (Section 1.1 Taxonomy) at the end of the Introduction section to explain the differences among those three concepts. The paper has been revised strictly following those definitions.
>
> (4) Regarding the table to illustrate the different attributes that can be generated from different baselines.
> - We added a table in Section 4.2.2, labeled as Table 2, to indicate which set of attributes can be generated from which model, including all the baselines and our DiffuseSG.
>
> (5) Regarding the diagram for Relation-ControlNet.
> - We added a figure in Section 3.2.2, labeled as Figure 3, to illustrate the Relation-ControlNet.
>
> (6) Regarding the results of DiffuseSG with different input representations (scalar, one-hot) with the IoU loss.
> - We have revised the “Ablations” paragraph in Section 4.2.3 and added the results of DiffuseSG with different input representations (scalar, one-hot) with the IoU loss into Table 4 (ablations) in Section 4.2.3. The conclusion is that the binary-bit encoding performs the best among the three encoding representations (binary-bit, scalar, and one-hot), both in the settings with and without the IoU loss, both on the Visual Genome and COCO-Stuff datasets. Pairing binary-bit encoding with the IoU loss gives the best performance.
>
> (7) Regarding the inputs to the LayoutDiffusion model.
> - When pairing DiffuseSG with LayoutDiffusion, the generated relation labels are not fed into LayoutDiffusion. Only the object labels and object bounding box locations are inputted. This motivates us to build Relation-ControlNet, where all the generated attributes from DiffuseSG can be utilized. However, as indicated in (the current) Table 6, LayoutDiffusion is still better than our Relation-ControlNet in terms of the evaluation scores. This is due to the fact that in the VG dataset, many relations can be inferred from bounding box positions. Given the better performance of LayoutDiffusion, we use it as our main conditional image generator for experiments and visualization.

---

> > ### Comment · Reviewer_nn1a · 2025-06-01
> > **Concerns well addressed**
> >
> > Thank you for the response, additional results, and revisions. My previous concerns and questions were well addressed. From my perspective, the quality of the paper has significantly improved as a result of these updates.

---

### Review · Reviewer_o2mt · 2025-05-14

**Summary Of Contributions:**

DiffuseSG presents a novel approach by integrating diffusion models for the joint generation of scene graphs and images. Unlike traditional methods, DiffuseSG can generate scene graphs unconditionally from noise, capturing both the structure and attributes effectively. Key contribution of this work:
- Joint modeling: Simultaneously generates scene graphs and corresponding images, ensuring consistency between the two modalities.
- Graph Transformer Denoiser: Employs a graph transformer as the denoising network within the diffusion process, handling both continuous and discrete attributes
- IoU Regularization: Introduces IOU regularization to enhance the spatial coherence of generated objects.

**Audience:**

Yes

**Broader Impact Concerns:**

I didn't observe any potential ethical concerns or negative societal implications associated with the proposed method.

**Claims And Evidence:**

Yes

**Requested Changes:**

**Major comments**

Based on my comments in the Weaknesses section, I encourage the authors to strengthen the motivation for why unconditional scene graph generation deserves more attention. While I suggested considering a conditional scene graph generation formulation as a possible alternative, I acknowledge that this may not align with the intended scope or contributions of the paper. Instead, I recommend that the authors more clearly describe the value and potential of the unconditional approach. Clarifying this motivation will help readers better appreciate the relevance and impact of this research direction.

**Minor Comments**

#1. In Section 3.1.2, the authors write:
 “Following Karras et al. (2022), we reparameterize the score function by a denoising function which maps the noise-corrupted data back to the clean data.” I suggest citing an earlier foundational work on this idea: Denoising Score Matching by Pascal Vincent (2010). While Karras et al. (2022) builds on this, Vincent’s work is more appropriate for introducing the original concept.

#2. Please clarify how the total variation difference is computed. Specifically, how is the L1 loss between p^(x)\widehat{p}(x)p​(x) and q^(x)\widehat{q}(x)q​(x) calculated?

#3. One key component of this joint model is the scene graph-guided image generation described in Section 3.2. Could you provide a performance comparison between your method (e.g., LayoutDiffusion and Relation-ControlNet) and previous works such as SceneGenie and SGDiff?

**Strengths And Weaknesses:**

**Strengths**

To enhance image fidelity and provide greater control in generative pipelines, this joint modeling approach is compelling, which is similar to DALL·E 2’s two-stage decomposition (which uses image embeddings instead of scene graphs). Modeling a complex marginal distribution through this factorization is a principled direction.

A core contribution of this work is the unconditional scene graph generation, composed of a graph transformer denoiser combined with IoU regularization. This regularization explicitly models object locations, which has not been adequately addressed in prior works.

**Weaknesses**

My main concern with this work is the practical effectiveness of the unconditional scene graph generation component. Scene graph-guided image generation is a promising direction for improving controllability in generative models, and many recent works have explored this avenue. However, unconditional scene graph generation remains less studied. I believe this is partly because scene graph generation itself is somewhat difficult to apply in many practical settings.

To strengthen the motivation of this work, could the authors consider conditional scene graph generation instead of the unconditional formulation? Specifically, rather than modeling the joint distribution P(S,I), it may be more effective to model P(S,I | C)=P(I | S,C)P(S | C), where C represents a class label, text description, or any other form of conditioning. If it can be demonstrated that explicitly incorporating a scene graph in this conditional framework improves the fidelity of image generation, or even if the fidelity remains similar but generation controllability improves, this would provide a much stronger and more practical motivation. Furthermore, such a formulation could be beneficial beyond generative modeling, such as in data augmentation for discriminative models, making it more broadly applicable than the purely unconditional approach.

---

> ### Author Response · Authors · 2025-05-28
> **To Reviewer o2mt**
>
> Dear Reviewer o2mt,
>
> Thank you for your review and feedback. Below we address each of your comments in turn.
>
> (1) Regarding the motivation of our paper.
> - We would like to address this point from two perspectives.
> - **First**, the ultimate goal we are trying to do is generating grounded scene graph and image pairs unconditionally via diffusion models. We have revised the second paragraph in the Introduction to better reflect the motivation of generating the pairs. The motivations are also pasted below.
> - “The benefits of such generative modeling would be multifaceted. First, it can be used to generate synthetic training data to augment training of discriminative grounded scene graph detection approaches discussed above. Second, it can serve as a generative scene prior which can be tasked with visualizing likely configurations of objects in the scene conditioned on partial observations via diffusion guidance. For example, where is the likely position of the chair given placement of the table and sofa. Third, it can be used for controlled image generation, by allowing users to first sample grounded scene graphs and edit them, and then conditioned on them, generate corresponding images.”
> - **Second**, because our main contribution is in generating the grounded scene graph unconditionally via a diffusion model. One more motivation of doing so is that, the diffusion model should be able to correct a counterintuitive grounded scene graph. That is, given the diffusion model a noised version of a grounded scene graph containing counterintuitive information, after denoising, in the cleaned scene graph, the counterintuitiveness should be fixed.
>
> (2) Regarding citing Denoising Score Matching by Pascal Vincent (2010).
> - We have now cited the paper in Section 3.1.2.
>
> (3) Regarding the clarification of the triplet total variation difference measure.
> - We have revised the relevant descriptions in the paper to make this more clear. The revised descriptions are pasted below.
> - “We use the total variation difference (TV) to measure the marginal distribution difference between the generated triplet labels and the ground-truth ones. Specifically, assuming that the generated empirical distribution is $\hat{p}$ with the ground-truth being $\hat{q}$, where $\hat{p}$ and $\hat{q}$ are vectors sized as the number of unique triplets combining all generated and ground-truth triplets. The TV is calculated as $\frac{1}{2} |\hat{p} - \hat{q}$|. Note, if a triplet does not exist in either $\hat{p}$ or $\hat{q}$, then its relevant entry in $\hat{p}$ or $\hat{q}$ is $0$.”
>
> (4) Regarding comparing between (LayoutDiffusion and Relation-ControlNet) and (SceneGenie and SGDiff).
> -  **About SceneGenie.** We can not find SceneGenie’s code online. Thus we are unable to compare it in terms of performance given the short time period. But generally speaking, SceneGenie could also be one of the conditional image generator candidates because it can generate images guided by text (incorporating object and relation labels) and bounding box locations. SceneGenie is also built upon Stable Diffusion.
> -  **About SGDiff.** SGDiff accepts only object and relation labels as input for image generation, and thus it may not be able to generate the objects at the specific locations when conducting our evaluation, which is specifying the object locations to calculate classification accuracies. This means that our evaluation may not suit SGDiff. SGDiff has code and checkpoint available online, and thus we are still able to calculate some quantitative numbers for SGDiff.
> - We take the code and the checkpoint available at the SGDiff's Official Github Repository to conduct experiments, specifically, calculating the quantitative numbers in the setting of Table 6 (grounded scene graph classification evaluation). The results of DiffuseSG along with other models are listed below.
> - Table 6: **Grounded scene graph classification evaluation results** under the PredCls and SGCls settings.
> |-Method-  |  -PredCls Mean Triplet Acc ↑-  |  -SGCls Object Acc ↑-  |  -SGCls Mean Triplet Acc ↑-  |  -FID ↓-  |
> |:------------|:------------:|:------------:|:------------:|:------------:|
> |Ground-Truth Images | 30.65 | 70.31 | 15.77 | 0.0|
> |LayoutDiffusion | **27.85** | **56.79** | **10.04** | 15.73|
> |Object-ControlNet | 26.68 | 55.05 | 8.35 | **15.32**|
> |Relation-ControlNet | 27.08 | 47.94 | 9.07 | 15.99|
> |SGDiff | 19.92 | 17.85 | 1.15 | 27.25|
> - From the results, we see that the SGDiff model that we evaluated is worse than all of LayoutDiffusion, Object-ControlNet, and Relation-ControlNet. We believe the reasons are twofold. First, it generates images with lower perceptual quality, indicated by a higher FID score. Second, since it can not take object locations as input while generating images, it may not be able to render objects and relations at the appropriate locations that we want. Because our evaluation setting may not suit SGDiff, we decide to not put its results into our paper.

---

> > ### Comment · Reviewer_o2mt · 2025-06-05
> >
> > Thank you for your detailed response and thoughtful revision. My original concerns were primarily about the motivation for unconditional scene graph generation and the lack of sufficient comparison to prior work. The latter has been adequately addressed in your response.
> >
> > However, the motivation remains somewhat unconvincing to me. While the arguments presented are reasonable, they are not clearly reflected or supported by the experiments in the current manuscript. That said, I do believe the revised version shows clear improvement over the previous one, and overall, I find the work to be meaningful.

---

### Author Response · Authors · 2025-05-28
**To All Reviewers**

Dear Reviewers,

Thank you all for the careful reviews!

We have uploaded the revised paper based on all of your reviews. The revised parts are highlighted in color blue. Given the ambiguity of the term “scene graph”, in all of our responses, we are now using the term “grounded scene graph” to describe the visual scene representation containing object labels and bounding box locations, and relation labels. That is, our DiffuseSG model generates grounded scene graphs unconditionally.

---

### Decision · Action_Editor_gtFs · 2025-07-06

**Recommendation:** Accept as is

**Additional Comments:**

All three reviewers acknowledge that the paper presents an interesting approach for joint generation of scene graphs and images by leveraging a diffusion-based denoising model in scene graph generation tasks. The main technical contribution is its design that generates scene graphs unconditionally from noise using a graph transformer denoiser and integrates a bounding box IoU penalty term. The majority of the reviewers considered that the paper is well-written with clear motivation, and the evaluation is comprehensive, covering both scene graph generation and various downstream tasks and demonstrating strong empirical performance in scene graph and layout generation.

However, the reviewers also raised several concerns initially regarding: 1) unclear motivation for unconditional scene graph generation; 2) missing comparisons to related scene graph-guided image generation methods; 3) limited support for technical contributions in scene graph-conditioned image generation; and 4) insufficient illustrations of the training scheme, certain network design aspects, and the process of conditional image generation.

The author's response provided a thorough revision of the paper, including further clarification, more illustrations, and additional comparisons to the related works. The reviewers found their initial concerns on 2) and 4) satisfactorily addressed but remained unconvinced on 1) and 3) due to insufficient evidence from experiments and method design, respectively. Overall, all three reviewers considered that those limitations are outweighed by the core contributions of this work and reached a consensus of positive recommendations (2 Leaning Accept and 1 Accept). The AE concurs with the reviewers' assessment — the work presents an effective diffusion-based scene graph generation method, which is a valuable contribution and supported by clear empirical evidence, and hence recommends acceptance. The authors are encouraged to incorporate further evidence or clarification on the unresolved concerns raised by the reviewers in the final version.

**Audience:**

Yes

**Audience Explanation:**

The topic of scene graph generation is of interest to the audience in machine learning and computer vision.

**Claims And Evidence:**

Yes

**Claims Explanation:**

The paper presents a novel framework for joint grounded scene graph-image generation, primarily focusing on unconditional scene graph generation. To this end, it introduces a new diffusion model, DiffuseSG, which integrates a graph transformer as a denoiser and an IoU-based regularization term for generating heterogeneous node and edge attributes in scene graphs. The experiments demonstrate strong empirical performance in scene graph generation and a range of downstream tasks such as conditional image generation, scene graph completion, and synthetic data generation to enhance scene graph detection.

The revised paper after discussion phase received unanimous approval from all three reviewers. Overall, the work is well-motivated, clearly written, and includes a thorough empirical study that supports its main claims.